# LymphoAtlas: a dynamic and integrated phosphoproteomic resource of TCR signaling in primary T cells reveals ITSN2 as a regulator of effector functions

Marie Locard-Paulet[1,†,‡] (iD), Guillaume Voisinne[2,†], Carine Froment[1], Marisa Goncalves Menoita[2], Youcef Ounoughene[2,3], Laura Girard[2,3], Claude Gregoire[2], Daiki Mori[2,3], Manuel Martinez[3], Hervé Luche[3], Jerôme Garin[4], Marie Malissen[2,3], Odile Burlet-Schiltz[1], Bernard Malissen[2,3] (iD), Anne Gonzalez de Peredo[1,*] (iD) & Romain Roncagalli[2,**] (iD)

## Abstract

T-cell receptor (TCR) ligation-mediated protein phosphorylation regulates the activation, cellular responses, and fates of T cells. Here, we used time-resolved high-resolution phosphoproteomics to identify, quantify, and characterize the phosphorylation dynamics of thousands of phosphorylation sites in primary T cells during the first 10 min after TCR stimulation. Bioinformatic analysis of the data revealed a coherent orchestration of biological processes underlying T-cell activation. In particular, functional modules associated with cytoskeletal remodeling, transcription, translation, and metabolic processes were mobilized within seconds after TCR engagement. Among proteins whose phosphorylation was regulated by TCR stimulation, we demonstrated, using a fast-track gene inactivation approach in primary lymphocytes, that the ITSN2 adaptor protein regulated T-cell effector functions. This resource, called LymphoAtlas, represents an integrated pipeline to further decipher the organization of the signaling network encoding T-cell activation. LymphoAtlas is accessible to the community at: https://bmm-lab.github.io/LymphoAtlas.

**Keywords** dynamic biological processes; ITSN2; LymphoAtlas; phosphoproteomics; TCR signaling network

**Subject Categories** Immunology; Proteomics

**Mol Syst Biol. (2020) 16: e9524**

## Introduction

Cell-surface receptors constantly detect and integrate signals emanating from the extracellular environment. In most cases, the conversion of extracellular stimuli into intracellular cues is encoded by post-translational modifications (PTMs) of proteins occurring rapidly after receptor engagement. Among PTMs, phosphorylations propagate signaling by transiently regulating enzymatic activities, protein localizations, and protein–protein interactions (PPIs). Such signaling cascades play a major role in T cells, whose functionalities depend on signals delivered by the T-cell receptor (TCR; Brownlie & Zamoyska, 2013). In contrast to other receptors, which bind a predetermined ligand, the TCR recognizes a variety of major histocompatibility complex-associated antigenic peptides (pMHC), thereby encoding extracellular signals into an adaptive immune response. TCR stimulation also conditions signaling emanating from auxiliary T-cell sensors such as co-stimulatory or cytokine receptors. Thus, the TCR is a master regulator of T-cell functions and its dysfunction can lead to immune disorders (Fischer et al, 2015).

Given the essential role of the TCR, tremendous efforts have been made over the last decades to map and characterize the molecular mechanisms responsible for signal propagation following its engagement. The TCR is associated with CD3 chains that bear immunoreceptor tyrosine-based activation motifs (ITAMs) providing the entry point for downstream signal propagation (Chakraborty & Weiss, 2014). Upon TCR engagement, signaling is initiated by the phosphorylation of these ITAMs by the tyrosine kinase LCK. Doubly phosphorylated ITAMs provide docking sites for the ZAP70 kinase,

1 Institut de Pharmacologie et de Biologie Structurale (IPBS), Université de Toulouse, CNRS, UPS, Toulouse, France
2 Centre d'Immunologie de Marseille-Luminy, INSERM, CNRS, Aix Marseille Université, Marseille, France
3 Centre d'Immunophénomique, INSERM, CNRS UMR, Aix Marseille Université, Marseille, France
4 CEA, BIG, Biologie à Grande Echelle, INSERM, U1038, Université Grenoble-Alpes, Grenoble, France
   *Corresponding author. Tel: +33 (0) 561175541; E-mail: gonzalez@ipbs.fr
   **Corresponding author. Tel: +33 (0) 491269478; E-mail: roncagalli@ciml.univ-mrs.fr
   †These authors contributed equally to this work
   ‡Present address: Novo Nordisk Foundation Center for Protein Research, University of Copenhagen, Copenhagen, Denmark

which in turn becomes tyrosine-phosphorylated and activated. Those initial TCR signaling steps are followed by further downstream molecular events involving the recruitment of key adaptors such as LAT and SLP-76 allowing signal diversification through the mobilization of additional proteins with diverse enzymatic activities, ultimately promoting the acquisition of T-cell effector functions (Balagopalan *et al*, 2010; Gaud *et al*, 2018; Voisinne *et al*, 2019).

Studies analyzing human immunodeficiencies, immortalized T-cell lines, or genetically engineered mouse models have identified key proteins that are required for proper T-cell activation along with important PTMs that can regulate the enzymatic activities of kinases, phosphatases or transcription factors. Despite these efforts, a global understanding of this complex signaling network is still lacking. Indeed, conventional approaches have often focused on a particular component of the TCR network and have not allowed a broad monitoring of the proximal or more distal parts of the signaling pathways mobilized by TCR stimulation. Furthermore, most available data do not include time-resolved quantitative measurements. Finally, the heterogeneity of the experimental models used between different studies makes it difficult to obtain a coherent and global description of the TCR pathway and to predict the effects of genetic or drug perturbations in primary cells.

Recent advances in mass spectrometry (MS) analysis promise to greatly improve our understanding of complex biological networks by allowing the identification of large sets of PPIs, and the quantitative comparison of proteomes and phosphoproteomes in a hypothesis-free fashion (Aebersold & Mann, 2016). Such large-scale MS-based approaches have been already used in a few studies to characterize protein phosphorylation and associated molecular mechanisms in primary T cells. Navarro *et al* (2011) applied SILAC protein metabolic labeling on murine P14 cytotoxic T lymphocytes (CTLs) to analyze phosphorylation following long-term (1-h) stimulation of the TCR with its cognate peptide, and identified around 2,000 phosphorylated peptides, among which 22% were TCR-regulated. Subsequently, to study the mechanisms of PKD2, a kinase important for effector cytokine production after TCR engagement, a similar strategy was implemented to compare the phosphoproteomes of wild-type and PKD2-deficient CTLs after 5 min of TCR activation (Navarro *et al*, 2014). This work led to a more extensive coverage of 15,000 site-specific phosphorylations in antigen receptor-activated CTLs, although no comparison was performed with unstimulated cells, precluding the identification of TCR-regulated phosphorylation sites. In a recent study, Joshi *et al* (2017) also used phosphoproteomics to analyze regulatory T-cell (Treg) suppression mechanisms on primary human conventional T cells, upon TCR stimulation and Treg-mediated suppression, respectively. Using a coculture system and a quantitative approach based on isotopic dimethyl labeling of peptides, the authors could detect around 2,000 phosphopeptides and quantify around 1,000 of them in three different T-cell states (unstimulated, TCR-stimulated with anti-CD3/anti-CD28 antibodies, and Treg-suppressed). These studies, based either on metabolic protein labeling or on dimethyl peptide labeling, were limited in the number of conditions or time points that could be included and compared. In addition, they focused on the global phosphoproteome, composed mainly of phosphorylated serine and threonine sites. To overcome this limitation, Ruperez *et al* (2012) introduced an additional step of purification to specifically enrich phosphorylated tyrosine residues. Using this approach, the authors identified a total of 2,883 phosphorylated peptides in CD4 human T cells stimulated for 5 min with anti-CD3 antibodies, including 48 peptides phosphorylated on tyrosines. Altogether, the development of these methods paved the way for in-depth analysis of signaling and phosphorylation induced during T-cell activation.

Here, our aim was to apply such unbiased, large-scale MS-based methods to provide a detailed and comprehensive picture of the basic mechanisms involving protein phosphorylation, in the first minutes following TCR activation in murine primary CD4$^+$ T cells. We used a label-free quantitative method, allowing to include several time points and replicates in our experimental setup, to quantify and monitor the phosphorylation dynamics of residues during the first 10 min after TCR stimulation. The use of a modern, fast-sequencing Orbitrap MS instrument enabled us to mine the phosphoproteome at a depth of 13,000 unique phosphorylated peptides and around 7,000 phosphorylation sites with localization confidence. By including an additional step of enrichment of rare phosphotyrosine (pY)-containing peptides, we were able to quantify a large collection of pY sites (> 250) in primary T cells. In an effort to provide a useful descriptive resource, we performed a thorough computational analysis of this time-resolved data set and also developed a Web interface for easy visualization of phosphosites kinetics. The analysis of phosphorylation time courses allowed us to identify TCR-regulated phosphosites along with their different dynamic patterns. It highlighted a rapid mobilization of molecular components involved in cytoskeleton remodeling, transcription, and translation processes. Our data illustrate the phosphorylation dynamics of the well-known upstream players of the TCR signaling pathway, as well as that of novel components. To follow-up on one of them, we used a fast-track gene extinction approach based on the CRISPR/Cas9 system in primary mouse T cells, and we demonstrated that the intersectin 2 (ITSN2) adaptor protein regulates T-cell effector functions by controlling TCR surface down-regulation upon antigenic stimulation.

## Results

### Mapping phosphorylation sites in primary T cells prior and over the first 10 min following TCR stimulation

To generate an extensive data set of phosphorylation events taking place in primary T lymphocytes, CD4$^+$ T cells from wild-type mice were purified and briefly expanded by stimulating them with anti-CD3 plus anti-CD28 antibodies *in vitro* (see Materials and Methods). After 2 days in culture supplemented with interleukin 2 (IL-2), effector T cells enter a state of quiescence characterized by their inability to further divide and proliferate in the absence of new TCR stimulation. At that stage, TCR engagement by anti-CD3 plus anti-CD4 antibodies leads to a transient increase in global tyrosine and serine/threonine phosphorylation, as shown by the analysis of total protein lysates immunoblotted with phospho-specific antibodies (Fig EV1A). To capture the molecular events induced by TCR engagement in a hypothesis-free and site-specific fashion, briefly expanded T cells (as described above) were left unstimulated or stimulated via their TCR for 15, 30, 120, 300, or 600 s prior to phosphoproteomic analysis (Fig 1A). Four

independent experiments were performed in which phosphopeptides were enriched using TiO$_2$ beads and analyzed by LC-MS. In three of these experiments, an additional enrichment step using phosphotyrosine immunopurification (pY-IP) was introduced to achieve a better mapping of pY peptides. Therefore, the TiO$_2$ data set and the pY-IP data set correspond, respectively, to 24 and 18 independent samples, each analyzed in triplicate nano-LC-MS runs. These led to the identification of 13,009 unique phosphorylated peptides corresponding to 9,702 and 560 phosphosites—or combination of sites in the case of multiple phosphorylated peptides—in the TiO$_2$ and the pY-IP data set, respectively (Fig EV1B). From these peptides, we were able after filtering (PEP value > 0.01, localization score > 75%, elimination of sites with high number of missing values; see Materials and Methods) to relatively quantify 7,180 phosphorylation sites or combinations of sites (Dataset EV1). These correspond to 6,984 unique, non-redundant sites (Fig 1B and C), including 5,659 phosphoserines (pS), 1,070 phosphothreonines (pT), and 254 pY present on 2,118 proteins in primary T cells. The analysis of the number of phosphosites present per protein indicated that although some proteins were phosphorylated on more than ten residues (*e.g.,* ACIN1, ADD3), 77% of phosphoproteins displayed between one and three phosphosites (Fig EV1C). We checked whether the probability of detection of phosphopeptides could be affected by the corresponding protein abundance or protein length. We found no relationship between the length of proteins and the number of detected phosphosites per protein (Fig EV1D). In order to evaluate the protein relative abundances in CD4$^+$ T cells, we used a proteome generated using SDS–PAGE pre-fractionation (Voisinne *et al*, 2019) and extracted the iBAQ values (sum of all the peptide intensities divided by the number of observable peptides of a protein) for each detected protein. Among the 2,181 unique proteins for which we quantified at least one phosphorylation site, we could quantify the abundance of 1,871 proteins (86%) in the proteome. Comparison of this proteomic data set with our phosphoproteomic data suggested that at least 29% of the proteins expressed in T cells were phosphorylated and that the detection of phosphoproteins was not biased toward highly abundant molecules (Fig EV1E and F). Moreover, our analytical workflow allowed us to reproducibly track rare and transient molecular modifications in primary T cells. In particular, the pY enrichment allowed the proportion of phosphorylated tyrosines to reach nearly 4% in our data set, therefore constituting, to our knowledge, the largest collection of pY sites identified in primary mouse T cells. The analysis of annotations (UniProt keywords) within the set of identified

phosphoproteins in the context of the CD4$^+$ T cell proteome revealed an enrichment of proteins involved in transcriptional regulation and cytoskeleton remodeling and also showed an association with other PTMs such as ubiquitination, methylation, and isopeptide bond formation (Fig EV1G). Interestingly, a large fraction of phosphosites were already detected in the unstimulated condition (89% of detected phosphoproteins) and constitute a resource *per se* since some of these phosphoproteins could be required to maintain cell integrity and protein expression and stability. For example, multiple phosphorylations were detected on the ARFGEF1/2 proteins involved in vesicular trafficking and maintenance of the Golgi structure. As highlighted by annotations, high occurrences of constitutive phosphorylations were also observed on transcriptional repressors (NCOR1/2, SIN3A, HDAC1/4/5) involved in methylation and histone deacetylation leading to the formation of repressive chromatin structures. This data set contains more than 1,000 phosphosites that were not previously reported in the literature or in the dedicated database PhosphoSitePlus (www.phosphosite.org), accounting for 14% of our global data set (Fig 1B and C). Collectively, these results constitute an extensive and quantitative resource mapping phosphosites specific to primary CD4$^+$ T cells.

### Identification of TCR-regulated phosphorylation sites

We next sought to identify sites whose phosphorylation was regulated by TCR stimulation. With this aim, we set up a workflow based on label-free relative quantification of intensities across all time points, followed by an ANOVA-based statistical analysis (see Materials and Methods). In brief, the phosphosites presenting well-resolved kinetics (a minimum of two biological replicates with three common time points) were considered regulated when they presented an absolute fold change ≥ 1.75 associated with a corrected *P*-value ≤ 0.05 between two time points (Dataset EV1). With this approach, we identified 731 phosphorylation sites (or combination of sites) that were regulated in the course of TCR stimulation, including 611 pS, 77 pT, and 52 pY (Fig 2A and B). These regulated sites constituted around 10% of the phosphosites of the total data set and mapped to 492 regulated phosphoproteins (*e.g.,* around 7% of all the proteins expressed in T cells, according to the proteome described above) (Fig EV2A). A similar strategy applied to the data obtained from the analysis of the peptides before phosphoenrichment allowed us to quantify more than 80% of the abundance of regulated phosphoproteins. Importantly, these data indicate that protein abundances were not impacted by TCR

▶

**Figure 1. Description of the phosphoproteomic data set and its generation.**

A  Experimental workflow to quantify phosphorylations occurring at the early time points after TCR stimulation. Digested peptides were analyzed by MS before further enrichment to control the impact of TCR stimulation on protein abundances (proteome). Further steps of TiO$_2$ enrichment and phosphotyrosine immunoprecipitation (pY-IP) were conducted to quantify changes in the phosphorylated serines and threonines (pST phosphoproteome), as well as phosphorylated tyrosines (pY phosphoproteome), respectively (see the Materials and Methods section for more details). Schematic representations of the measured phosphokinetics are presented on the right.

B, C  Phosphosites identified after TiO$_2$ enrichment (B) and phosphotyrosine immunoprecipitation (pY-IP) (C) Left panel: pie chart presenting the proportion of identified phosphorylated serines (pS), threonines (pT), and tyrosines (pY) in the two data sets. Right panel: cumulative number of unique identified phosphosites across the biological replicates (3 and 4 independent experiments for the TiO$_2$ and pY-IP, respectively). Phosphorylated serines, threonines, and tyrosines are presented individually next to the total number of identified sites (pSTY). The phosphosites that were not previously reported in any species in the PhosphoSitePlus database (www.phosphosite.org) are reported in isolation in the graph on the right.

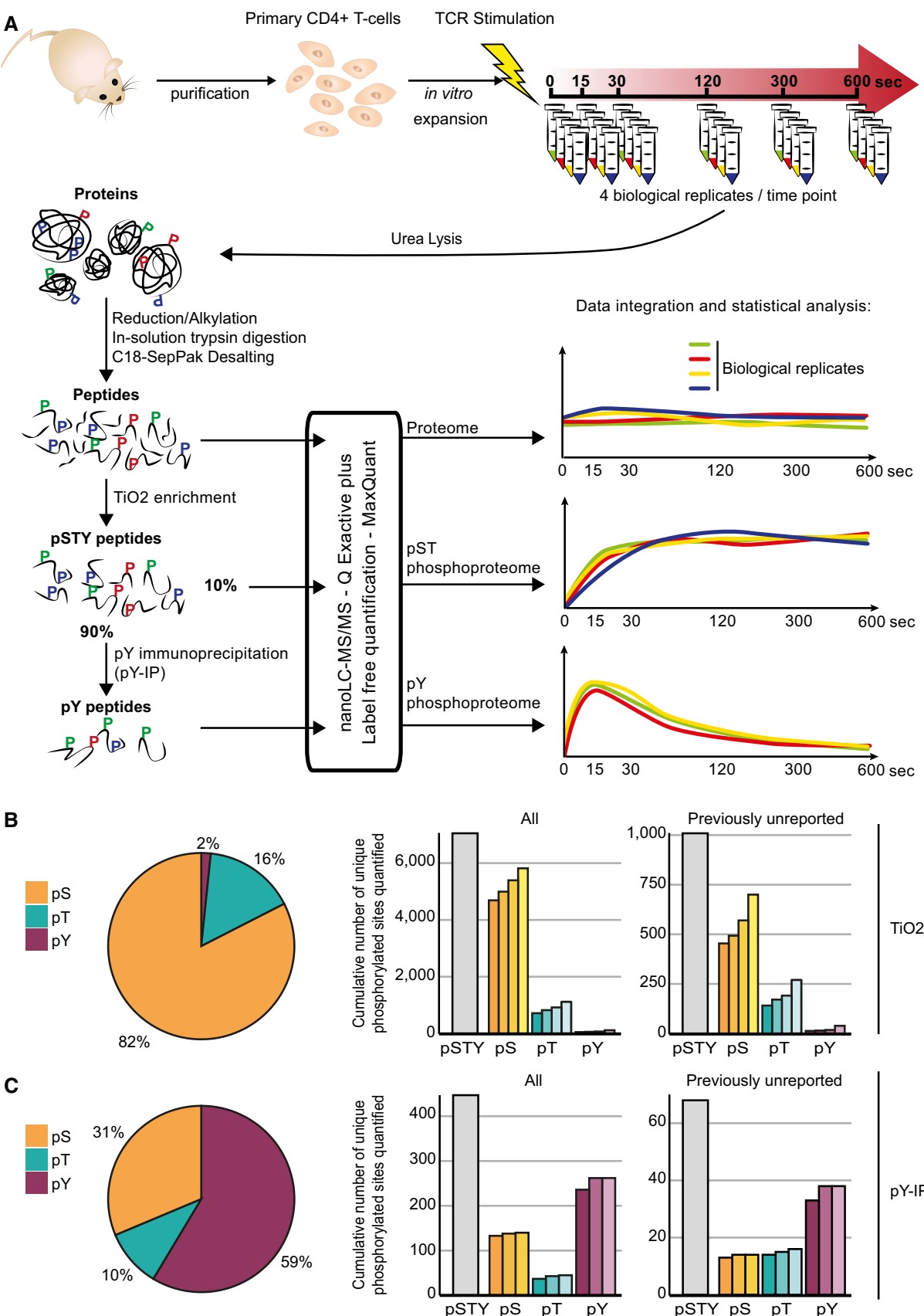

**Figure 1.**

activation at these early time points (Figs 1A and EV2B, and Dataset EV2) and that changes in phosphopeptide quantities directly reflected phosphorylation or dephosphorylation events. The analysis of annotations (UniProt keywords) enriched within the set of regulated phosphosites compared to the set of phosphosites of the total data set revealed that proteins associated with cytoskeletal organization ("Cytoskeleton", "Actin-binding", and "Synapse"), ion transport ("Calcium transport", "Chloride channel") and with specific protein domains ("SH3 domain", "SH2 domain") were impacted by TCR stimulation (Fig EV2C). Many major actors of the canonical TCR signaling pathway such as the TCR complex itself, the kinases LCK and ZAP70, the phospholipase C-gamma-1 (PLCg1), and the adaptors SLP76 and LAT also contained TCR-regulated phosphosites (Fig 2C). Regulated phosphosites also covered distal and ubiquitous pathways associated with cellular activation as illustrated by the identification of components of the MAPK signaling pathway (RAF1, BRAF, ERK1, ERK2, RSK2, and MAPK14) and of the calcium signaling pathway (ITPR3, STIM1/2). Regulation of these pathways through TCR activation is known to trigger active actin remodeling, leading to bulk membrane movements, morphological changes, and formation of the immunological synapse. Consistently with the analysis of annotations, important phosphorylation changes were detected on a large set of proteins involved in cytoskeleton remodeling, such as LCP-1, VASP, Cofilin-1, or STMN. In addition, TCR stimulation regulated the phosphorylation of sites on many proteins involved in transcriptional regulation, including important factors reported to control the gene expression program associated with T-cell responses (HDAC7, NFATC1/2, FOXO3, BACH2). TCR stimulation also affected the phosphorylation of plasma membrane-anchored proteins such as surface receptors (CCR7, leukosialin/CD43, LY9, CD97, and CD5) or the transmembrane molecule LIME1 providing additional docking sites for the recruitment of downstream effectors. Finally, TCR-regulated phosphosites were identified on proteins known to negatively regulate the TCR signaling pathway, such as the ubiquitin ligases CBL and CBL-B, phosphatases (PTPN1, PTPN6, PTPN22, PTPN7, UBASH3A), and the signal-transducing adapter molecule 2 (STAM2). Noticeably, among the 730 phosphorylation sites that we identified as significantly regulated according to the criteria described above, some exhibited very strong regulation, with fold changes of high amplitude (shown as a list in Dataset EV1). Additionally, Fig 2C shows a schematic representation of phosphosites found to be significantly regulated following TCR stimulation in this study, and known from the literature to be functionally related to this pathway (also see Dataset EV1). It covers many of the major previously reported regulators of T-cell activation.

## Chronological orchestration of biological processes underlying T-cell activation

To tackle the complexity of the time-resolved data set and extract additional biological insights, we sought to apply t-distributed stochastic neighbor embedding (t-SNE) to transform the multidimensional data (MS intensity per time points after stimulation for each phosphosite) into a two-dimensional plot. Here, each point corresponds to a regulated phosphosite, which is located in the neighborhood of those with similar phosphorylation dynamics (Fig 2D). Within the t-SNE map, we identified 13 clusters containing 24 to 111 phosphosites using an approach based on maximum local densities (Rodriguez & Laio, 2014; Fig 2C and D). The variety of phosphokinetics displayed by these clusters revealed diverse TCR-dependent phosphoregulations. The first two clusters (C1, 76 sites, and C2, 73 sites) displayed a very early and transient phosphorylation increase peaking at 15 or 30 s, respectively, followed by a rapid decrease. Similarly, clusters C3 (44 sites) and C4 (25 sites) displayed a rapid increase in phosphorylation levels also peaking at 15 or 30 s, but remained sustained (C3) or even quite stable (C4) for these groups. Other clusters showing increased phosphorylation (C5–C8) were characterized by a delayed phosphorylation peak. Indeed, maximum intensities of phosphorylation of C5 and C6 were at 120 s and were followed by either down-regulation (C5) or maintenance (C6). For C7 and C8, we observed a gradual and continuous amplification of the phosphorylation signal all along the time course, which remained sustained (C7) or was even increasing (C8) at the later time points. Although most of the sites were up-regulated upon TCR engagement (73 % of them fell in clusters C1–C8), our analysis also revealed five clusters (C9–C13) displaying decreased phosphorylation upon TCR stimulation. Of note, while tyrosine residues were mainly involved in relatively persistent phosphorylation events (clusters C3, C6, and C7), serine and threonine residues exhibited a wider range of dynamic patterns with many examples of quick and transient phosphorylation (C1-2), progressive and sustained modification (C8), or dephosphorylation (C9-13; Fig EV3A).

Analysis of the annotations enriched within the set of phosphosites in a given cluster when compared to the set of all TCR-regulated phosphosites revealed that clusters with very early and transient phosphoregulations (C1 and C2) were enriched with sites corresponding to proteins associated with "GTPase activation", "Actin-binding", "Synaptosome", and "Cell projection", several terms that characterize the cytoskeletal remodeling process, a biological event required for T-cell activation (Ritter *et al*, 2015; Fig 3A and B). For example, quick and transient serine phosphorylations were observed within the EVH2 domains of EVL (S257,

---

**Figure 2. Identification and dynamics of TCR-regulated phosphorylation sites.**

A, B　Output of the statistical analyses of the TiO$_2$-fractionated (A) and the pY-IP-fractionated (B) data sets. Left panel: number of phosphosites significantly regulated upon TCR stimulation presented in bar plots for each time point, next to the total number of regulated sites across the entire time course (total). The number of phosphorylated threonines (pT) and serines (pS) regulated in the TiO$_2$ data set is indicated in yellow and green, respectively. Right panel: representative volcano plot presenting the statistical significance distribution against the log$_2$-transformed fold change between 300 s and the unstimulated control. For each condition, phosphosites were considered significantly up-regulated (red) or down-regulated (blue) when displaying a corrected *P*-value ≤ 0.05 (ANOVA test) and an absolute fold change ≥ 1.75 (see Materials and Methods for more detailed information).

C　　Mapping of a subset of TCR-regulated phosphosites on the canonical TCR signaling network. Phosphosites are color-coded according to their kinetic of phosphorylation, as presented in (D).

D　　t-SNE view and clustering analysis of phospho-regulated sites. The 13 clusters were defined using local density within the t-SNE plot (see Materials and Methods).

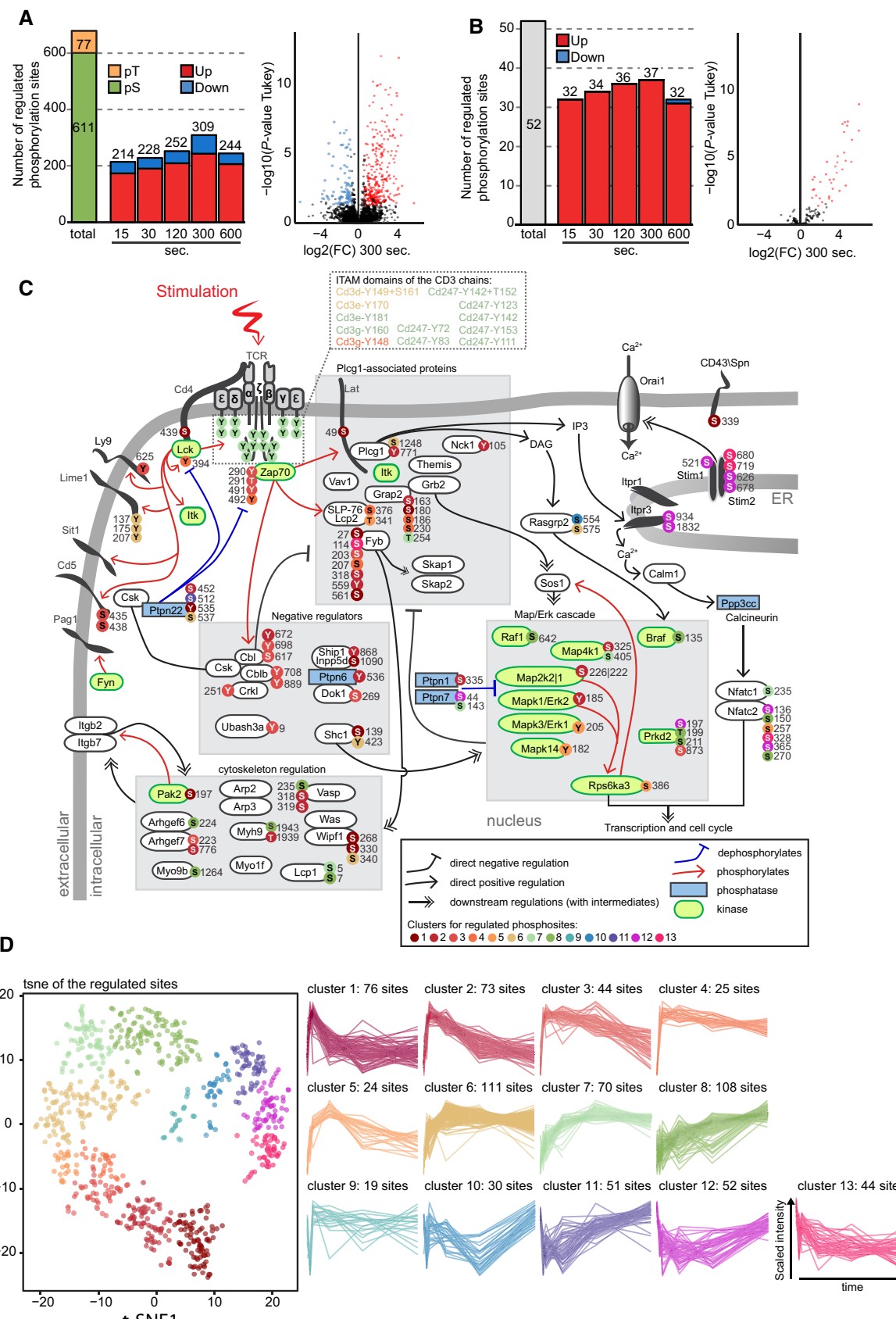

**Figure 2.**

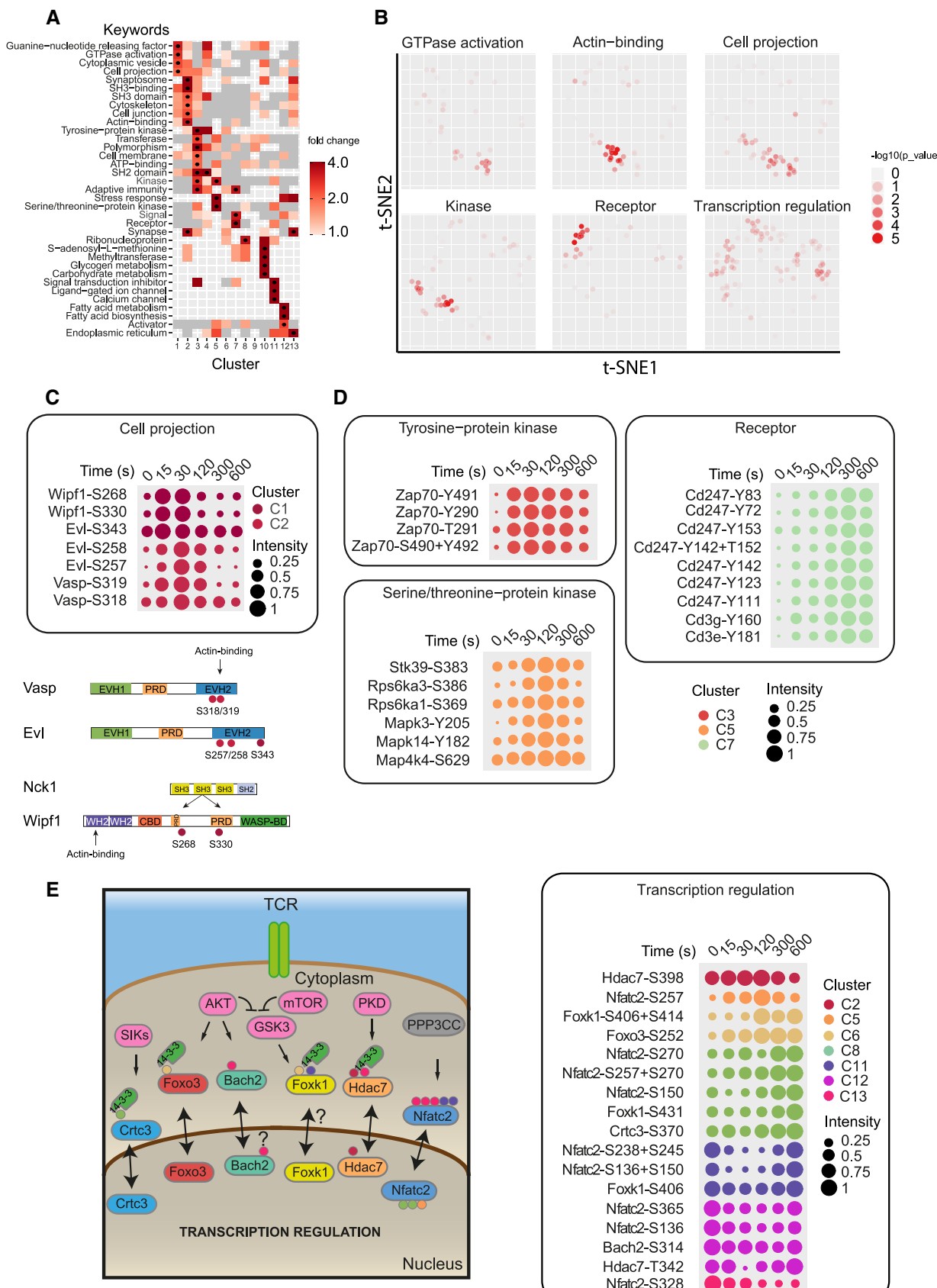

Figure 3.

**Figure 3.  Enrichment and dynamics of cellular processes triggered by early TCR signals.**

A   Heat map showing UniProt keywords (https://www.uniprot.org/) enriched in each cluster. Black dots indicate clusters with significantly enriched keyword terms (hypergeometric test P-value ≤ 0.01, fold change ≥ 2, number of annotated sites ≥ 2; gray squares: fold change < 1).

B   t-SNE plot highlighting phosphosites associated with the keyword terms specified above each plot. Dot transparency is scaled according to the P-value corresponding to the local enrichment of the annotation term (hypergeometric test, see Materials and Methods for more detailed information).

C   Dot plot showing dynamics of scaled intensities over time of selected phosphosites associated with the keyword "Cell projection" (dots color-coded per cluster). The linear structure of the selected proteins with the corresponding regulated phosphosites is depicted below.

D   Similar representation as in (C) for the terms "Tyrosine-protein kinase", "Serine/threonine-protein kinase", and "Receptor".

E   Illustration of the transcription factors for which activation through nucleus-cytoplasmic shuttling is regulated by phosphorylation. The normalized dynamics/intensities of the corresponding phosphosites are indicated on the right panel (as in C, D).

S258, S343) and VASP (S318, S319) (Fig 3C). VASP EVH2 domain (310-TTLPRMKSSSSVTTS-324) contains actin-binding sites, and phosphorylation of its serine residues has been shown to promote actin polymerization (Doppler *et al*, 2013). We also detected coincidental phosphorylation of the WAS/WASL-interacting protein family member 1 (WIPF1) within (S330) or adjacent (S268) to the binding sites of NCK second SH3 domain (Fig 3C). This protein–protein interaction is known to contribute to the release of the auto-inhibited WASP conformation, which, in turn, activates Arp2/3 to induce actin filament assembly (Donnelly *et al*, 2013; Rohatgi *et al*, 2001). The functional enrichment analysis also highlighted the accumulation of phosphosites on protein kinases in clusters C3–C5 (Fig 3A and B), particularly the crucial phosphosites characterizing the *trans*-phosphorylation enzymatic activity of ZAP70 (Y491 and Y492), which reached a maximum of phosphorylation between 15 and 30 s and were progressively dephosphorylated after 120 s (Fig 3D). Finally, the keyword "Receptor" was enriched in cluster C7, due to the over-representation of sites from phosphorylated ITAMs of the CD3 chains. This revealed that phosphorylation of the TCR complex homogeneously increased up to 5 min and remained persistent ten minutes after initial TCR engagement despite the attenuation of the ZAP70 kinase activity (Fig 3D). Whether early and late CD3 chain phosphorylations trigger distinct or similar downstream signals remains to be determined.

### Early TCR signals regulate the transcriptional and translational machinery

In contrast to the terms analyzed above, which were enriched in specific kinetic clusters, other biological processes were associated with proteins presenting phosphosites more randomly distributed in the t-SNE. This was the case for sites related to "Transcription regulation" (Fig 3B), which indicates that the phosphoregulation events underlying this process were diverse and took place throughout the 10-min TCR stimulation time course analyzed here. Their heterogeneous dynamics highlight the various molecular mechanisms that control transcription. One example is the polycomb-repressive complex 1 (PRC1), which promotes transcription through chromatin remodeling. Several phosphorylation events were detected on the different components of this complex (Phc3-T607, Cbx4-S350, Rnf2-S41), each with different kinetics (Fig EV3B). TCR stimulation induced the progressive phosphorylation of Rnf2-S41, an event preventing histone H2A ubiquitination and allowing its acetylation, thereby promoting transcription (Niessen *et al*, 2009; Rao *et al*, 2009). We also observed the early dephosphorylation of Phc3-T607 and Cbx4-S350 which suggests that these non-previously reported events could also regulate the function of this complex. Another

consequence of the phosphorylation affecting transcriptional regulators is to promote their localization in specific subcellular compartments. For example, protein phosphorylation on serine residues can induce their interaction with 14-3-3 proteins and result in their nuclear exclusion (Brunet *et al*, 2002). Accordingly, we identified phosphorylation events on factors previously reported to be regulated by such a mechanism (Fig 3E), like the CREB-regulated transcription coactivator 3 (Crtc3-S370), the forkhead box protein O3 (Foxo3-S252), and the forkhead/winged-helix family k1 (Foxk1-S406/S414/S431), all showing serine phosphosites regulated upon TCR stimulation (Clark *et al*, 2012; He *et al*, 2018; Singh *et al*, 2010). In particular, the increased phosphorylation characterizing the residue S252 localized in the NLS of FOXO3 indicates that the nuclear-to-cytosolic translocation and the 14-3-3 binding known to relieve the repression of mitogenic genes could occur < 10 min after TCR stimulation (Figs 3E and EV3C; Brunet *et al*, 1999).

A scattered t-SNE distribution was also observed for sites on proteins associated with the terms "Translation regulation" and "Protein biosynthesis" (Fig EV3D). Indeed, many subunits of complexes involved in the translation initiation (EIF3B, EIF4B, EIF4G1, EIF4G3, and EIF5B) and elongation (EEF1A1, EEF1B, EEF1D, and EEF2) were affected by distinct phosphorylation changes. However, these displayed synchronized phosphorylation dynamics within each protein complex (Fig EV3E). The elevated number of regulated phosphosites within these complexes, and the fact that only few of them (Eef2-T57, Eif4g3-S1150) have already been characterized, highlights our lack of knowledge regarding the complexity and rapidity of the molecular mechanisms that regulate essential cellular processes such as protein translation (Rose *et al*, 2009; Srivastava *et al*, 2012).

Overall, our analysis revealed the coherent orchestration of critical biological processes as early as a few minutes upon TCR engagement in primary mouse T lymphocytes.

### Profiling kinase activities in the course of TCR stimulation

To gain insights into the dynamics of kinase activities underlying TCR-mediated cell activation, we mapped kinase–substrate relationships from the PhosphoSitePlus database, which provides a data set containing experimentally determined substrates, sequences, cognate kinases, and metadata curated from the literature (www.phosphosite.org/). We restricted our analysis to kinases that could be identified in our proteome of CD4[+] T cells (Voisinne *et al*, 2019). Analysis of substrate enrichment per cluster revealed significant associations between activity of kinases and specific clusters, thereby reflecting the temporal dimension in which kinases operate (Fig 4A). For example, substrates corresponding to ERK and AKT

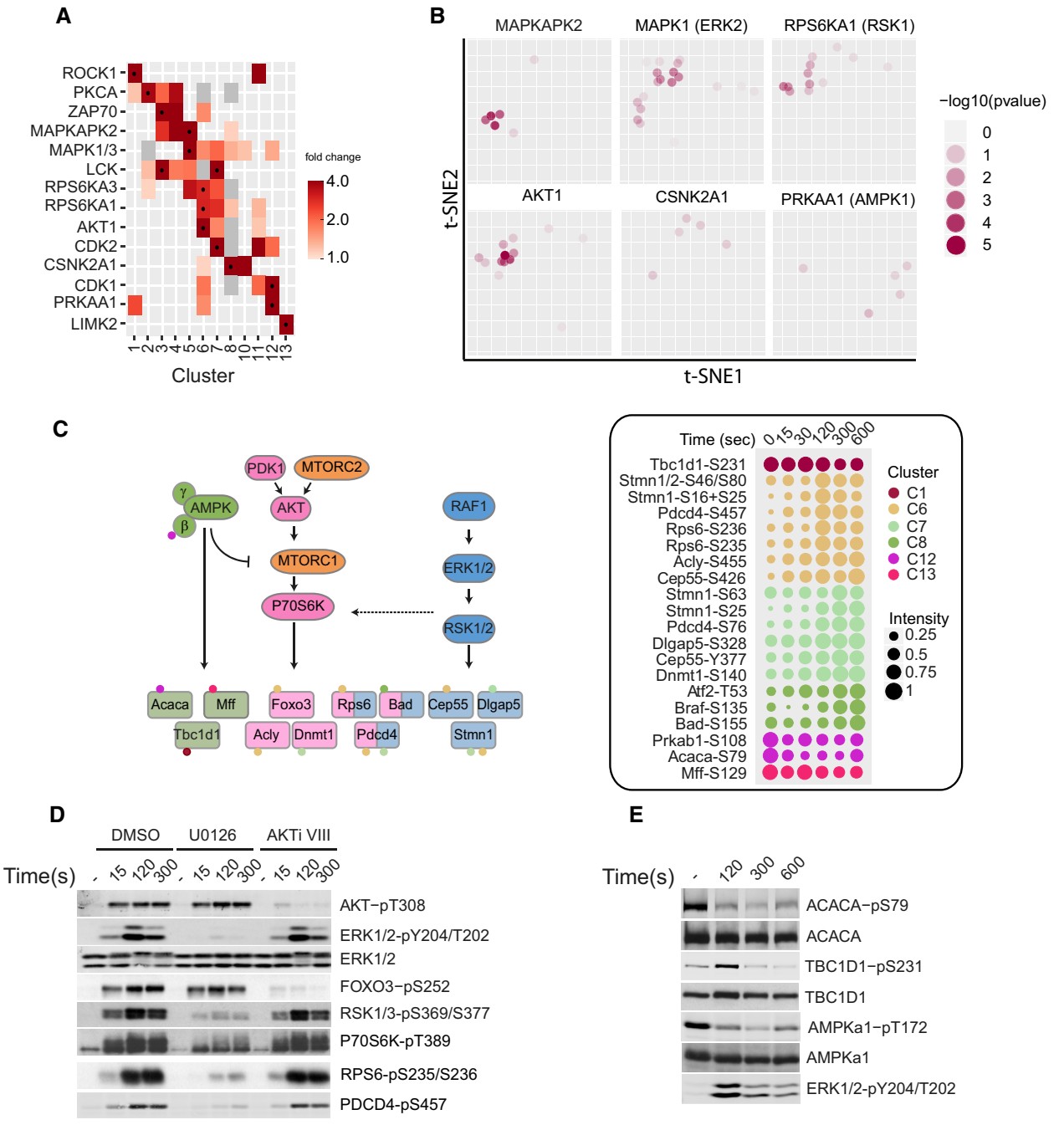

**Figure 4.　Dynamics of kinase activities.**

A　Heat map showing kinase activities based on enrichment of their specific substrates for each cluster. Black dots indicate clusters with significantly enriched kinase activity (hypergeometric test *P*-value ≤ 0.05, fold change ≥ 2, number of annotated sites ≥ 2; gray squares: fold change < 1).

B　t-SNE plot displaying local enrichment of phosphosites corresponding to known substrates associated with the specified kinase. Dot transparency is scaled according to the *P*-value corresponding to the local enrichment of the annotation term (hypergeometric test, see Materials and Methods for more detailed information).

C　Illustration of the kinase–substrate interconnections characterizing the AMPK, AKT, and ERK signaling pathways. Substrate phosphosites dynamics are displayed and color-coded according to their clusters, and the corresponding scaled kinetics are presented on the right.

D　Immunoblot analysis of equal amounts of proteins from total lysates of CD4[+] T cells treated with DMSO, U0126 (10 µM), or AKTi VIII (10 µM) and left unstimulated (−) or stimulated for 15, 120, or 300 s with anti-CD3 and anti-CD4 antibodies, probed with the indicated phospho-specific antibodies. Anti-ERK1/2 immunoblot served as a loading control.

E　Equal amounts of proteins from total lysates of CD4[+] T cells left unstimulated (−) or stimulated for 120, 300, or 600 s with anti-CD3 and anti-CD4 antibodies were analyzed by immunoblot with antibodies probing the indicated phosphosites/proteins.

Data information: In the phosphosite labels, "+" and "/" separate the phospholocalizations of doubly phosphorylated peptides or peptides having two proteins identifications with different phosphorylation localizations, respectively. Note that although TBC1D1−pS231 site falls into C1, its t-SNE coordinates are close to the clusters associated with dephosphorylation, a result consistent with its late dephosphorylation state.

kinases were preferentially enriched in clusters C5-C8 (Fig 4A–C). More specifically, we identified phosphorylation of STMN1, CEP55, DLGAP5 and ACLY, DNMT1, and FOXO3, known to be specific to the ERK and AKT signaling axis, respectively, although some of these kinase–substrate relationships have yet to be confirmed in primary T cells. Interestingly, phosphorylation of Pdcd4-S457, Bad-S155, and Rps6-S236/379 is indistinctly attributed to the AKT or the ERK axis in the literature and databases (Fig 4C). This observation can reflect a crosstalk between the two signaling pathways or alternatively, may be due to the diversity of cell types or engaged receptors analyzed and reported in public databases. To better characterize these sites and refine the specificity of the signals associated with TCR engagement, we treated T cells with specific MEK (U0126) or AKT1/2 (AKTi VIII) inhibitors and analyzed phosphorylation of their substrates (Fig 4D). As expected, phosphorylation on specific sites of MEK substrates such as ERK1/2-Y204/T202 and RSK1-S369/S377 was selectively inhibited by U0126, while treatment with AKTi VIII specifically reduced FOXO3-S252 and AKT-T308 phosphorylations. Interestingly, in these inhibitory conditions, phosphorylation of PDCD4-S457 was strongly reduced only by U0126, indicating a predominant dependence on the ERK signaling pathway for this site upon TCR activation in primary T cells (Fig 4D). PDCD4 phosphorylation on S457 has been shown to promote its degradation, thus preventing its translation inhibitory function. Therefore, this result suggests a rapid induction of translational processes within minutes of the TCR stimulation through PDCD4 and the ERK pathway. Similarly, we observed that the phosphorylation of RPS6 and P70S6K was strongly impaired upon inhibition of ERK but not of AKT. Although these substrates are mainly associated with the canonical AKT pathway, our results suggest that in the context of TCR signaling they could rather be regulated by ERK, potentially through RSK kinases (Roux et al, 2007).

In addition, we found that substrates of CDK1, PRKAA1, and LIMK2 were over-represented in dephosphorylated clusters C12 and C13 (Fig 4A and B). PRKAA1 (also AMPK1) is a central regulator of metabolism and energy balance in T cells, as in many other cell types. It negatively regulates mTORC1 signaling and promotes catabolic metabolism (Fig 4C). In addition, it has been shown that AMPK1 is required to generate effector memory responses upon a secondary antigen challenge (Rolf et al, 2013). We observed that TCR stimulation regulated the phosphorylation of AMPK1 substrates involved in processes that characterize proliferating T cells, such as fatty acid synthesis (acetyl-CoA Carboxylase, ACACA), glucose uptake (TBC1 domain family member 1, TBC1D1), or mitochondrial metabolism (mitochondrial fission factor, MFF; Fig 4C). Given that these were dephosphorylated, we hypothesized that AMPK1 activity was reduced following TCR stimulation and assessed it by probing T172 phosphorylation, which is known to reflect AMPK1 kinase activity (Hurley et al, 2005). The phosphorylation of T172 observed in quiescent cells progressively decreased upon stimulation, a result consistent with the dephosphorylation of the non-catalytic subunit β1 of AMPK1 (Prkab1−S108) identified by MS analysis (Fig 4C and E). Therefore, these results indicate that early TCR signaling represses AMPK1 enzymatic activity and suggest that T cells shift to aerobic glycolysis within minutes after TCR ligation.

### Intersectin 2 regulates T-cell effector functions

Proximity between phosphosites within the t-SNE plot reflects temporal coincidence of phosphorylation and can reveal molecular and functional relationships. By focusing on phosphotyrosine sites, we noticed that pY sites of STAM2 and ITSN2 were close to those of the CD3 chains (Fig 5A). In the case of the STAM2 phosphosite Stam2-Y371, this could be explained by the fact that this site is within an ITAM, possibly allowing its regulation by Src and ZAP70 kinases (Pandey et al, 2000). For the ITSN2 phosphosite (Itsn2-Y554), it could suggest an alternative functional relationship with the CD3 chains. ITSNs (ITSN1 and 2) are multi-modular scaffolding proteins, involved in the endocytosis pathway of various cell types (Tsyba et al, 2011). ITSNs are expressed under two splicing isoforms. The ITSN-short (ITSN-S) is composed of two Eps15 homology (EH) domains and a central coiled-coil domain followed by a series of SH3 domains. The longer form (ITSN-L) possesses an extended C-terminal part encoding for an exchange factor (DBL), a pleckstrin (PH), and a C2 domain. Although ITSN2 is ubiquitously expressed, the exploration of mouse public databases showed that it is preferentially expressed in B and T lymphocytes (www.immgen.org). In B cells, ITSN2 is required to promote germinal center formation and antibody production to influenza virus infection (Burbage et al, 2018). In addition, we also detected this protein in the interactomes of the CBL and CBL-B ubiquitin ligases in primary T cells, suggesting its involvement within the TCR signaling pathway (Voisinne et al, 2016).

**Figure 5. ITSN2 deficiency lowers TCR activation threshold.**

A  t-SNE plot highlighting regulated pY sites (for sake of space, only the gene names are reported). Phosphosites of STAM2 and ITSN2 are displayed in red.

B  Cas9-EGFP OT-I CD8⁺ T cells were transfected with control sgRNA (sgEGFP) or with two different sgRNA targeting *Itsn2* (sgITSN2-1 or sgITSN2-2). Transfected cells were stimulated for 48 h with N4 peptide MHC tetramers (0.1 nM) in the presence or absence of soluble anti-CD28 antibody or with IL-7 as control. Cell-surface expression of CD69 and CD5 was analyzed by flow cytometry (Z-score normalization of the geometric means of fluorescence MFI per experiment) in four independent experiments. Comparison between sgEGFP and sgITSN2 conditions was performed using a paired *t*-test. Box plot elements: Center line corresponds to median, box limits correspond to the first and third quartiles, whiskers indicate variability from Q1–1.5. IQR to Q3 + 1.5 IQR. Outliers are shown as black dots.

C  Proliferation of OT-I Cas9-EGFP CD8⁺ T cells transfected with EGFP or with ITSN2 sgRNA activated for 48 h with N4 peptide MHC tetramers (0.01–0.1 nM) in the presence or absence of soluble anti-CD28 antibody or with PMA and ionomycin (PMA/Iono) or with IL-7. Data are presented as mean ± SD from four proliferative measures of two independent sgEGFP and sgITSN2 transfections. Data are representative of three independent experiments.

D  IFN-γ production of cells treated as in (C) was assessed by intracellular staining after 24 h of stimulation.

E  Proliferation of similar cells as in (C) activated for 48 h *in vitro* with a range of increasing doses of N4 peptide MHC tetramers in the presence of soluble anti-CD28 antibody.

Data information: For panels (C) and (E): Data are representative of at least three independent experiments, and comparison between sgEGFP and sgITSN2 conditions was performed using a paired *t*-test (*$P \leq 0.05$; **$P \leq 0.01$; ***$P \leq 0.001$; NS, non-significant).

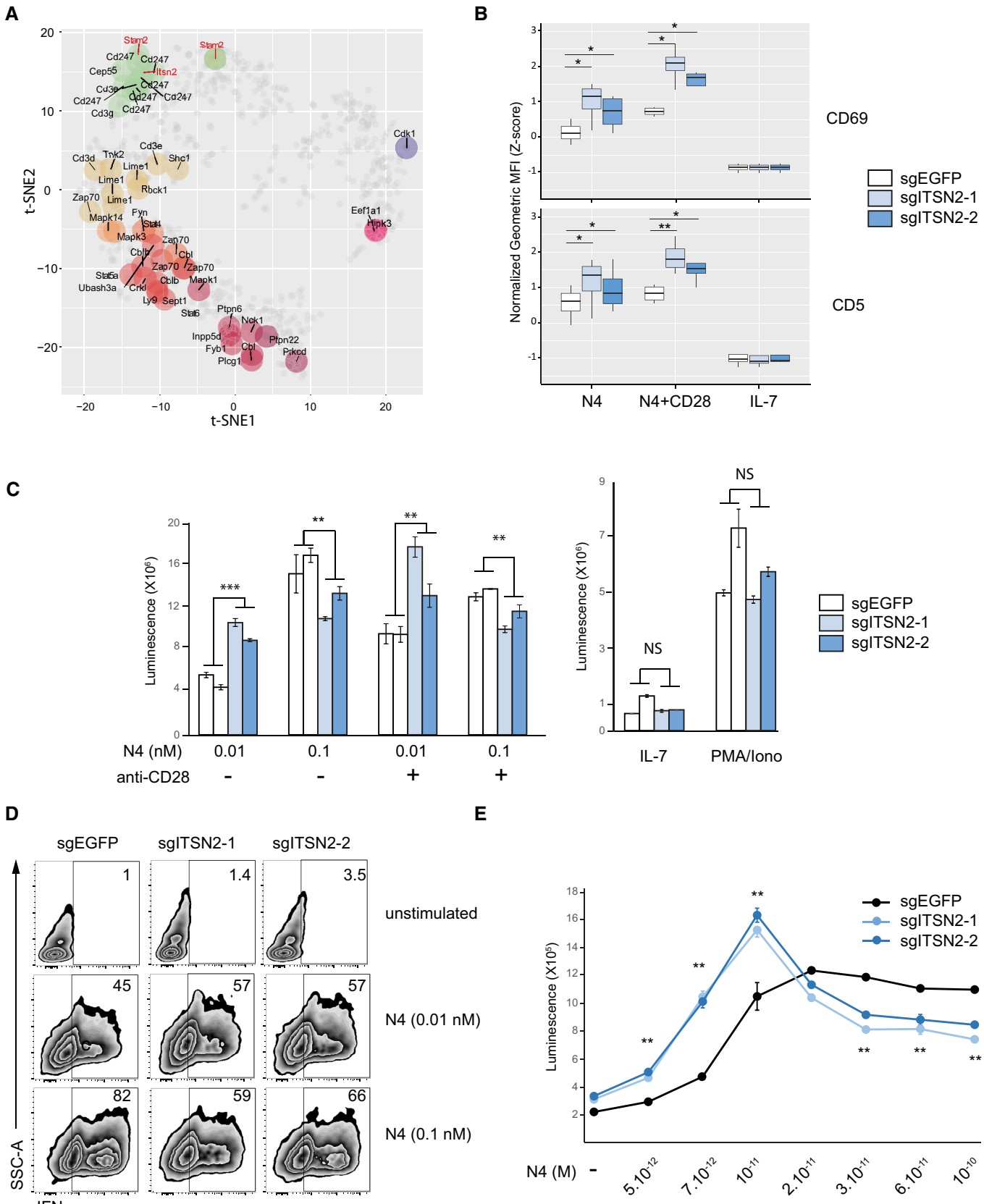

**Figure 5.**

To confirm that Itsn2-Y554 phosphorylation correlated with CD3 chain phosphorylations during TCR stimulation with a cognate antigen, we generated an independent phosphoproteome data set from OT-I CD8$^+$ T cells that express a TCR specific for the ovalbumin (OVA)-derived SIINFEKL (N4) peptide bound to H-2Kb MHC-I molecules. OT-I T cells were stimulated with pMHC-I tetramers laden with N4 peptide for 15, 30, 120, 300, and 600 s. Globally, analysis of this data set showed results very comparable to those obtained with CD4$^+$ T cells (Fig EV4, Dataset EV3). In brief, 59% of the phosphosites identified and 88% of the phosphosites regulated by TCR stimulation in the CD4$^+$ T cells were also detected in OT-I CD8$^+$ T cells (Fig EV4A–C). Moreover, phosphorylation kinetics of phosphosites regulated by TCR stimulation were remarkably conserved across both data sets (Fig EV4B and C). Notably, the phosphoproteins and their regulations we previously focused on were unambiguously confirmed by this data set (Fig EV4D). Importantly, OT-I-based data also confirmed that Itsn2-Y554 phosphorylation was regulated upon TCR activation with a dynamic similar to those of the CD3 chains, suggesting that the potential functional relationship between ITSN2 and the TCR observed in the CD4$^+$ T cells is preserved in CD8$^+$ T cells.

To investigate the role of ITSN2 in primary T cells, OT-I TCR transgenic mice were crossed onto constitutive Cas9-EGFP-expressing mice (Platt et al, 2014), and purified CD8$^+$ T cells from their progenies were transfected with specific sgRNA to selectively inactivate Egfp (control) or the Itsn2 gene. With EGFP sgRNA, this procedure routinely resulted in the fluorescent protein extinction in 80-85% of transfected cells (Fig EV5A). Accordingly, targeting Itsn2 with two different guides (sgITSN2-1 and sgITSN2-2) showed a drastic reduction in ITSN2 abundance compared with EGFP-targeted cells, indicating that one or both copies of the Itsn2 gene were knock-out in a large fraction of cells (Fig EV5B). These cells were stimulated with pMHC-I tetramers conjugated with the SIINFEKL (N4) peptide in the absence or presence of anti-CD28 antibody. The abundance of surface markers such as CD69, CD5, and CD44 significantly increased after 48 h of stimulation in Itsn2-targeted cells compared with control cells revealing an enhanced activation of cells deprived of the adaptor (Figs 5B and EV5C). Next, we assessed the impact of Itsn2 inactivation on proliferation and cytokine production upon TCR activation. Interestingly, stimulation of ITSN2-targeted cells exhibited contrasting phenotypes whether cells

were incubated with low or high doses of agonist peptide (0.01 nM or 0.1 nM). Indeed, the addition of a low concentration of tetramers induced increased proliferation and IFN-γ production of Itsn2-targeted cells compared with EGFP-targeted cells, while an opposite effect was observed at a high tetramer concentration (Fig 5C and D). To obtain a finer resolution, we stimulated cells with a range of increasing doses of pMHC-I tetramers and identified a breaking point between $10^{-11}$ and $2 \times 10^{-11}$ M of agonist peptide (Fig 5E). These results indicate that ITSN2-targeted cells have a lower activation threshold compared with wild-type cells. Additionally, reduced effector functions of Itsn2-targeted cells with high doses of peptides could result from an over-activation of induced cell death (AICD) mechanism (Green et al, 2003) or an exaggeration of the regulatory mechanisms responsible for the bell-shaped dose response observed in T cells (Rogers & Croft, 1999; Lever et al, 2016).

To explore the molecular mechanisms by which ITSN2 regulates T-cell functions, we stimulated Egfp- and Itsn2-targeted T cells with N4-laden pMHC-I tetramers and analyzed the phosphorylation of TCR downstream effectors by immunoblot and single-cell mass cytometry. Short-time stimulations induced a similar magnitude of total tyrosine phosphorylation and phosphorylation of ZAP70, CBL, AKT, and ERK molecules in both control and Itsn2-targeted cells (Fig 6A). Moreover, no significant difference was observed with the mass cytometry analysis covering a larger panel of proteins and phosphoproteins (Fig 6B and Appendix Fig S1A). The normal early signaling events but dysregulated proliferation and cytokine secretion in ITSN2-targeted T cells suggested that signaling differences could emerge on longer time scales. Since ITSN2 has been reported to regulate endocytosis, we tested the impact of its deficiency on cell-surface TCR levels (McGavin et al, 2001). Unstimulated sgITSN2-treated CD8$^+$ OT-I T cells displayed comparable amounts of surface TCR to those of control cells (Fig 6C and Appendix Fig S1B). However, incubation of cells with increasing doses of N4 peptide for 2 and 4 h showed a reduced TCR down-modulation in ITSN2-targeted cells compared with control cells (Fig 6C and Appendix Fig S1B). Accordingly, further analysis of phosphoproteins by mass and flow cytometry after a prolonged stimulation revealed a sustained and enhanced activity of different signaling pathways in ITSN2-targeted T cells, with a predominant impact on molecules involved in the ERK1/2 signaling axis (Fig 6D and Appendix Fig S1C). Altogether, these results demonstrate that ITSN2

**Figure 6. Reduced TCR down-modulation in ITSN2-targeted cells.**

A   Immunoblot analysis of equal amounts of proteins from total lysates of Cas9-EGFP OT-I CD8$^+$ T cells transfected with sgEGFP, sgITSN2-1, or sgITSN2-2 and left unstimulated (−) or stimulated for 2 or 5 min with N4 tetramers (20 nM). Upper panel: membrane probed with antibody to phosphorylated tyrosine (anti-p-Tyr) or anti-LAT (loading control). Lower panel: membrane probed with antibodies to anti-CBL-pY774, anti-ZAP70-pY319/352, anti-AKT-pT308, anti-ERK1/2-pY204/T202, or anti-ERK1/2 (loading control).

B   Heat map showing normalized levels (z-score) of indicated surface markers and protein phosphorylations (left margin) from Cas9-EGFP OT-I CD8$^+$ T cells transfected with sgEGFP or sgITSN2-1 left unstimulated (−) or stimulated for 1, 3, or 10 min with N4 tetramers (1 or 10 nM) and analyzed by mass cytometry. Z-scores were calculated from hyperbolic arcsine (arcsinh)-transformed intensities. Results obtained from independent transfections with sgITSN2-1 and sgITSN2-2 are compiled in Fig EV5A.

C   Similar cells as in (A) stimulated with 1 nM of N4 peptide for the indicated time points (left panel) or stimulated with increased doses of N4 peptides for 4 h (right panel) were analyzed by FACS for surface TCR expression. Data are presented as mean ± SEM, and comparisons were performed using a paired t-test (*$P \leq 0.05$, **$P \leq 0.01$; ***$P \leq 0.001$). A representative analysis of three independent experiments is shown.

D   Heat map showing scaled expression levels (z-score) of indicated surface markers and protein phosphorylations (left margin) from Cas9-EGFP OT-I CD8$^+$ T cells transfected with sgEGFP or sgITSN2-1 and stimulated for 6 h with increasing doses of N4 peptides or PMA/ionomycin (PI) analyzed by single-cell mass cytometry (left). Z-scores were calculated from hyperbolic arcsine (arcsinh)-transformed intensities. Signal difference (mean arcsinh difference) between sgEGFP and sgITSN2-1 transfected cells is depicted on the right panel.

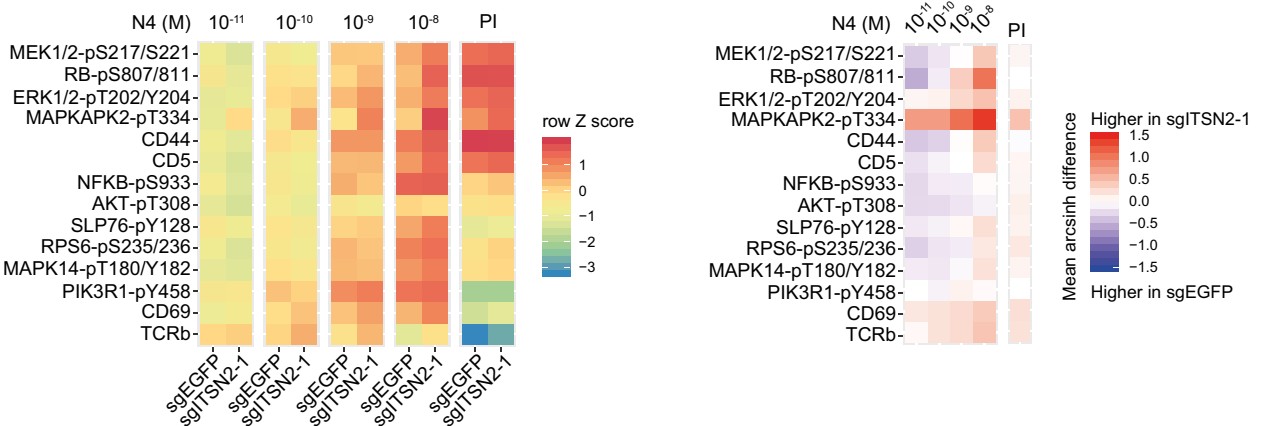

Figure 6.

plays a critical role in the regulation of the T-cell effector functions by controlling stimulation-dependent TCR internalization.

## Discussion

Pioneering phosphoproteomic studies have identified molecular events induced by receptor ligation (Nguyen *et al*, 2009; Brockmeyer *et al*, 2011; Humphrey *et al*, 2015; Navarro *et al*, 2011). LymphoAtlas complements these studies by focusing for the first time on early phosphorylation events triggered by receptor ligation in primary cells. This study provides a time-resolved, quantitative atlas of protein phosphorylations before and during the first 10 min of TCR stimulation. We quantified and determined the relative changes in phosphorylation on 7,180 sites (unique sites plus combination of sites quantified together on the same peptides) corresponding to more than 2,000 proteins. A number of these phosphosites (1,777 out of 7,180) were not reported in previous studies, thus highlighting the deep coverage of this data set. By specifically enriching pY-bearing peptides, we quantified the phosphorylation of 349 phosphotyrosine sites, thereby providing the largest collection of such sites in primary T cells. Analyzing phosphorylation kinetics in the course of TCR stimulation, we identified 731 phosphosites that significantly varied upon TCR engagement.

Analysis of our time-resolved data set revealed a coordinated and chronological induction of essential biological processes that constitute the first steps of T-cell activation. In particular, it suggested a more rapid mobilization of the effectors involved in the cytoskeleton remodeling, transcription, translation, and metabolic processes than initially thought. For example, the phosphorylation of Foxo3-S252 involved in its nuclear export (Brunet *et al*, 1999) was induced as early as 15 s after TCR stimulation. Interestingly, although we observed that tyrosine residue phosphorylations were mainly transient, very early events such as the phosphorylation of the TCR complex CD3 chains persisted over the entire course of stimulation. The kinase activities of ZAP70 and LCK/FYN began to decrease 2 min after stimulation, which could support the model where binding of ZAP70 to the TCR ITAMs prevents their access to phosphatases, and their dephosphorylation (Labadia *et al*, 1996; Sjolin-Goodfellow *et al*, 2015).

One of the most challenging issues in studies generating high-throughput data is to validate relevant biological hypothesis inferred by computational approaches. Here, using a fast-track CRISPR/Cas9 gene-editing system in primary cells, we assessed the function of selected components of the TCR signaling network in a particular transgenic background and independently of potential effects of gene deletion during T-cell development and selection. Such an approach was conducted for the *Itsn2* gene and revealed that the corresponding coding protein plays an essential role through regulating TCR down-modulation upon engagement. Consistently, prolonged TCR expression resulted in sustained TCR signaling, as demonstrated by enhanced activation of the ERK1/2 pathway. Although the precise mechanism by which ITSN2 regulates TCR surface expression remains to be determined, several prospective lines can be investigated. We previously showed that ITSN2 constitutively interacts with CBL and in a TCR-inducible manner with CBL-B (Voisinne *et al*, 2016). Among interaction partners shared by CBL and CBL-B, other proteins with similar interaction dynamics, such as EPS15L1 and SH3KBP1 (also CIN85), have been reported as

molecular partners of ITSN molecules (Nikolaienko *et al*, 2009; Wong *et al*, 2012). Interestingly, all these proteins are components of the endocytic machinery present in various cell types. Thus, it has been proposed that ITSNs could act as stabilizing scaffolds for the recruitment of endocytic proteins to the surface receptor activation sites. Such molecular cooperativity has been illustrated with ITSN1 that promotes CBL ubiquitin ligase activity, thereby enhancing EGFR ubiquitination and internalization (Martin *et al*, 2006; Okur *et al*, 2012). In lymphocytes, past studies have shown that CBL and CBL-B have redundant functions in peripheral T cells and are required to promote ligand-induced TCR down-modulation (Naramura *et al*, 2002). Therefore, it is possible that a similar mechanism to the one at play during EGFR internalization involving the formation of endocytic vesicles composed of EPS15L1, SH3KBP1, CBLs, and ITSN2 might account for TCR down-modulation at the T-cell surface following TCR engagement.

LymphoAtlas is a unique resource to explore the phosphorylation events underlying TCR signaling initiation in primary T cells. To facilitate the exploration of this data set by the scientific community, a Web application that interactively displays phosphorylation kinetics of selected sites from the LymphoAtlas database is available at https://bmm-lab.github.io/LymphoAtlas/ (see also Appendix Fig S2). By quantitatively capturing changes in phosphorylation dynamics, LymphoAtlas provides a rationale to the identification of therapeutic targets to advance immunotherapies based on T-cell reprogramming. Finally, the approach developed here lays the foundation for further systematic explorations of dysregulated signaling networks underlying diseases or resulting from molecular perturbations or gene inactivation in genetically engineered mice.

## Materials and Methods

### Mice

Wild-type C57B6/L, OT-I, and *Cas9-EGFP*-expressing Gt(ROSA) 26Sor[tm1.1(CAG-cas9*,-EGFP)Fezh] (Platt *et al*, 2014) mice were maintained in specific pathogen-free conditions, and all experiments were done in accordance with institutional committees and French and European guidelines for animal care.

### Flow cytometry

Stained cells were analyzed using an LSRII system (BD Biosciences). Data were analyzed with the Diva software (BD Biosciences). Cell viability was evaluated using SYTOX Blue (Life Technologies). The following antibodies were used: anti-CD5 (53–7.3), anti-CD4 (RM4-5), anti-CD8a (53–6.7), anti-TCRb (H57-597), anti-CD44 (IM7), and anti-CD69 (H1.2F3), all from BD Biosciences.

### T-cell isolation and short-term expansion

CD4[+] or Cas9-EGFP OT-I CD8[+] T cells were purified from pooled lymph nodes and spleens with a Dynabeads Untouched Mouse CD4[+] or CD8[+] T Cell Kit (Life Technologies); cell purity was 95%. CD4[+]-purified T cells were expanded for 48 h with plate-bound anti-CD3 (145-2C11, 5 μg/ml) and soluble anti-CD28 (37-51; 1 μg/ml both from EXBIO). Then, T cells were harvested and grown in

the presence of IL-2 (5–10 U/ml) for 48 h prior to stimulation for phosphoproteomic analysis.

### T-cell transfection with sgRNA

Cas9-EGFP OT-I CD8$^+$ T cells were purified from pooled lymph nodes and spleens with a Dynabeads Untouched Mouse CD8$^+$ T Cell Kit (Life Technologies); cell purity was 95%. Purified T cells were expanded for 48 h with plate-bound anti-CD3 (145-2C11, 5 μg/ml) and soluble anti-CD28 (37-51, 1 μg/ml; both from both from EXBIO). Then, $2 \times 10^6$ cells were transfected with 10 μg of specific sgRNAs (sgEGFP: GGGCGAGGAGCUGUUCACCG, sgITSN2-1: UAU CAGAUCUGAACAAGGAU, sgITSN2-2: GUUGUUUCAUGAUAGGAG GG) using Neon Transfection System (Invitrogen). Cells were kept in culture in the presence of IL-2 (5–10 U/ml) and IL-7 (5 ng/ml) for an additional 48 h prior to a subsequent experimental assay.

### T-cell proliferation and IFN-γ production

For proliferation assay, transfected Cas9-EGFP OT-I CD8$^+$ T cells were stimulated with various doses of SIINFEKL peptide MHC-I tetramers (provided by the NIH tetramer core facility) with or without soluble anti-CD28 (37-51; EXBIO) antibody. After 48 h of culture, T-cell proliferation was assessed with CellTiter-Glo® Luminescent (Promega). The resulting luminescence, which is proportional to the ATP content of the culture, was measured with a Victor 2 luminometer (Wallac, Perkin Elmer Life Science). For IFN-γ production, cells were stimulated with various doses of N4-laden pMHC-I tetramers with soluble anti-CD28 antibody for 24 h and treated with GolgiStop (BD Biosciences) for the last 4 h. After incubation, cells were washed and stained for dead cells, permeabilized (fixation/permeabilization buffer; Cytofix/CytoPerm BD Biosciences), and stained for intracellular IFN-γ (XMG1.2; BioLegend) before analysis by flow cytometry.

### TCR down-regulation assay

Transfected Cas9-EGFP OT-I CD8$^+$ T cells were stimulated with specified doses of SIINFEKL peptide (AnaSpec Inc.) for the indicated times. Then, cells were washed and stained for dead cells and with anti-TCRβ (H57−597) antibody and analyzed by flow cytometry.

### Stimulation of T cells for phosphoproteomic analysis

A total of $100 \times 10^6$ short-term expanded CD4$^+$ T cells from wild-type mice were left unstimulated or stimulated at 37°C with antibodies. In the latter case, CD4$^+$ T cells were washed in serum-free media, incubated with biotinylated anti-CD3 (0.2 μg per $10^6$ cells; 145-2C11) and anti-CD4 (0.2 μg per $10^6$ cells; GK1.5), and then cross-linked with purified streptavidin for the indicated times at 37°C. For OT-I CD8$^+$ T cells, stimulation was done with tetramers conjugated with the SIINFEKL (N4) peptide (20 nM; provided by the NIH tetramer core facility). Stimulation was stopped by snap-freezing the cells in liquid nitrogen.

### Sample preparation for proteomic analysis

Stimulated cells were thawed in urea buffer (8 M urea; 50 mM Tris–HCl pH 8; 1 mM sodium orthovanadate; PhosSTOP phosphatase inhibitor cocktail from Roche; cOmplete Mini EDTA-free from Roche) to reach a final urea concentration of 6 M, incubated for 15 min, and lysed with a Sonics-Vibracell before clarification by centrifugation for 30 min (11,860 g). Protein concentrations were determined using the Bio-Rad DC Protein Assay Kit, and total protein amounts were adjusted across samples within each experiment. Cysteine residues were reduced with 10 mM final of dithiothreitol for 1 h at room temperature, alkylated with iodoacetamide at a final concentration of 20 mM for 1 h in the dark at room temperature, and the alkylation was quenched with an additional 5 mM dithiothreitol for 30 min at room temperature. Urea concentration was reduced to 1 M by a sixfold dilution with 50 mM ammonium bicarbonate before trypsin digestion 1:100 (w:w trypsin: peptides) overnight at 37°C under agitation. Digested peptides were acidified using trifluoroacetic acid (TFA) 1% final, centrifuged for 30 min (4,696 g), and supernatants were desalted through C18 Sep-Pak cartridges (Waters) following manufacturer's instructions before being dried down under vacuum. An aliquot of 15 μg of these peptide samples was kept for triplicate MS analysis for protein relative quantification.

### Phosphopeptide enrichment

All phosphopeptide enrichments were performed with Titansphere TiO$_2$ beads (5 μm; GL Sciences 5020 75000), pre-washed in 80% MeCN, 1% TFA for 5 min, and then resuspended at 6 mg/ml in TiO$_2$ loading buffer (80% MeCN, 5% TFA, 1 M glycolic acid) for equilibration during 5 min. For each enrichment, the adequate volume of TiO$_2$ slurry necessary for a 6:1 (w:w) beads:peptide ratio was transferred into a new tube, spun down, and the supernatant eliminated. Peptide pellets were resuspended in TiO$_2$ loading buffer (1 μg/μl), and an equal amount of peptides for all samples was transferred into the tubes containing the conditioned TiO$_2$ beads for a 1-h incubation under agitation at room temperature. Beads were then sequentially washed with (i) loading buffer; (ii) 80% MeCN, 1% TFA; and (iii) 10% MeCN, 0.1% TFA. Phosphopeptides were eluted with 1% ammonium hydroxide (pH 11.3). A second elution was performed with 1% ammonium hydroxide, 40% MeCN (pH 11.3). The two eluates were pooled and filtered through an Empore cartridge C8-SD (4 mm/1 ml) before drying down under vacuum. 10% of these samples were subjected to MS analysis for phospho-serine/threonine relative quantification.

### Phosphotyrosine enrichment

90% of the TiO$_2$ eluates were further enriched for phosphotyrosine-containing peptides using PTMScan Phospho-Tyrosine Rabbit mAb Kits (P-Tyr-1000; Cell Signaling Technology) according to the manufacturer's instructions. Briefly, phosphopeptides were resuspended on ice in cold IAP buffer (14 mg/ml), sonicated for complete resuspension, and incubated with pre-washed PTMScan beads (80 μl of P-Tyr bead slurry for 20 mg of protein starting material) for incubation overnight at 4°C under mild agitation. Then, the beads were sequentially washed with IAP buffer and water before elution of the phosphopeptides with 0.15% TFA at room temperature for 10 min. Phosphotyrosine-containing peptides were then dried down under vacuum before MS analysis for phosphotyrosine relative quantification.

                                    *Molecular Systems Biology*   **16**: e9524 | 2020    **15 of 19**

## Nano-LC-MS/MS analysis

Peptides and phosphopeptides were analyzed by nano-LC-MS/MS using an UltiMate 3000 RSLCnano system (Dionex, Amsterdam, The Netherlands) coupled to a Q-Exactive Plus mass spectrometer (Thermo Scientific, Bremen, Germany). Dried samples were resuspended in 5% acetonitrile, 0.05% TFA spiked-in with iRT standard peptides (Biognosys) (1× final concentration of iRT in the case of pre-enriched peptides and $TiO_2$-enriched phosphopeptides; 0.1× final in the case of pY-enriched peptides). Pre-enriched peptides were resuspended at a theoretical concentration of 5 µg/µl according to initial protein assay, and 1 µl of sample was injected for each LC-MS analysis. Phospho-enriched peptides (pY and TiO2) were resuspended in 21 µl of solvent, and 5 µl of sample was injected for each LC-MS analysis. Samples were loaded onto a C18 precolumn (300 µm inner diameter × 5 mm, Dionex) at 20 µl/min in 5% acetonitrile, 0.05% TFA. After 5 min of desalting, the precolumn was switched online with a 75 µm inner diameter × 50 cm C18 column (in-house packed with Reprosil C18-AQ Pur 3 µm resin, Dr. Maisch; Proxeon Biosystems, Odense, Denmark) equilibrated in 95% solvent A (0.2% formic acid) and 5% solvent B (80% acetonitrile, 0.2% formic acid). The pre-enriched peptides and $TiO_2$-enriched phosphopeptide samples were eluted at 300 nl/min flow rate using a 240 min gradient of solvent B (4–12% in 44 min, 12–30% in 184 min, 30–40% in 12 min). The pY-enriched peptides were eluted at 300 nl/min flow rate using a 110-min gradient of solvent B (5% to 25% in 80 min, 25% to 50% in 30 min). The mass spectrometer was operated in data-dependent acquisition mode with the XCalibur software. The survey MS scans were acquired in the Orbitrap over m/z 400–1,600 with a resolution of 70,000 (at 200 m/z), an automatic gain control (AGC) target value of $1e^6$, and a maximum injection time of 200 ms. The 10 most intense ions per survey scan were selected at 3 m/z and fragmented by higher-energy collisional dissociation (HCD) using a normalized collision energy of 30. MS/MS scans were collected at 17,500 resolution with an AGC target value of $1e^5$ and a maximum injection time of 50 ms for peptide and $TiO_2$-enriched phosphopeptide samples, and an AGC target value of $1e^6$ and a maximum injection time of 250 ms for pY-enriched samples. A dynamic exclusion of 30 s was applied, and the "peptide match" option was set to "Preferred". For internal calibration, the m/z 445.120025 ion was used as lock mass.

## Protein identification and quantification

Raw MS files were analyzed by MaxQuant version 1.5.2.8. Data were searched with the Andromeda search engine against mouse entries of the SwissProt protein database (UniProtKB/Swiss-Prot Knowledgebase release2015_05, Mouse taxonomy, 16,699 entries), the Biognosys iRT peptides, and a list of potential contaminant sequences provided in MaxQuant 1.5.2.8. The search included methionine oxidation, protein N-terminal acetylation, and serine, threonine, and tyrosine phosphorylation as variable modifications, and carbamidomethylation of cysteines as a fixed modification. Specificity of trypsin digestion was set for cleavage after lysine or arginine, and three missed cleavages were allowed. The precursor mass tolerance was set to 20 ppm for the first search and 4.5 ppm for the main Andromeda database search, and the mass tolerance in MS/MS mode was set to 25 mmu. Validation was

performed through a false discovery rate set to 1% at protein and peptide-spectrum match (PSM) level determined by target-decoy search in MaxQuant (with additional filters for peptide validation: minimum length of 6 amino acids and a minimum Andromeda score of 40 for modified peptides). MaxQuant processing of files from phosphorylated peptides was performed independently for each enrichment step ($TiO_2$, pY-IP) and each biological series of experiment (replicates 1–4), with the "match between runs" option of MaxQuant enabled for label-free relative quantification of peptide ions across stimulation time points and injection replicates. The minimal ratio count was set to 1 for calculation of LFQ intensities.

## Experimental design and statistical rationale

Antibody-based TCR stimulation of primary mouse CD4[+] T cells was repeated four independent times with distinct pools of mice. For each condition (experiment, time point after stimulation, sample type), several MS injection replicates were performed, leading to 206 raw files considered for statistical analysis. The detailed description of each analysis (raw file name, sample type, biological replicate number, MS technical replicate number, analytical conditions) is available in the "Sample names" sheet of the Dataset EV1. N4-based TCR stimulation of OT1 CD8[+] T cells was repeated three independent times with distinct pools of mice. For each condition (experiment, time point after stimulation, sample type), three MS injection replicates were performed, leading to 108 raw files considered for statistical analysis. The detailed description of each analysis (raw file name, sample type, biological replicate number, MS technical replicate number, analytical conditions) is available in the "Sample names" sheet of the Dataset EV3. The quantitative proteomic analysis was performed using the statistical package R (R Development Core Team, 2012; http://www.R-project.org/) and R scripts related to the analysis can be found online: The antibody-based stimulation of CD4[+] T cells can be found on https://github.com/mlocardpaulet/TCR_ABStim2018; the N4 stimulation of OT1 CD8[+] T cells on https://zenodo.org with the https://doi.org/10.5281/zenodo.3556731; and the source code of the shiny-web application on https://github.com/mlocardpaulet/LymphoAtlas_App. Phosphorylation site relative quantification was performed independently on the $TiO_2$ and the phosphotyrosine IP samples with the intensities of the "Phospho (STY).txt" tables of MaxQuant. We treated independently the quantification values from mono- and multi-phosphorylated peptides in order to discriminate their regulation patterns upon TCR signaling. Intensities were first normalized for instrument variation using the MS intensities of the iRT spiked-in standards. Protein entries identified as potential contaminants by MaxQuant were eliminated from the analysis, as well as PSM with a PEP value > 0.01. $Log_2$-transformed values of technical replicates were averaged, and missing values were replaced as follows: If for a given condition (time point), a phosphorylation site was not quantified in any replicate experiment or quantified in only one of the replicates, missing values for that condition were imputed with a low-intensity value, obtained from a draw around the 5% quantile, and the standard deviation of the entire data set. At this stage, only the phosphosites with a localization score ≥ 75% in a minimum of one replicate were kept, and the phosphoserines and phosphothreonines were removed from the phosphotyrosine IP samples.

Conversely, the phosphotyrosines quantified in the pY-IP were removed from the $TiO_2$ data set. Moreover, for statistical analysis, phosphosites with high numbers of missing values were not considered: Only phosphorylation sites that were detected in at least 3 conditions (time points), and presenting in each of them a minimum of two quantification values from independent replicate experiments, were retained. For those, kinetics were normalized between biological replicates before being subjected to an ANOVA with Tukey's correction. To avoid biases coming from the random draw of replacement values, this statistical process was performed 200 independent times and phosphosites were considered statistically significant if they presented a p-value of the ANOVA $\leq 0.05$ and a minimum of one couple of time points with more than 2 points per condition and a corrected *P*-value $\leq 0.05$ with an associated absolute fold change $> 1.75$ in a minimum of 90% of the iterations. Protein relative quantification was performed in a similar way with the normalized LFQ intensities from the "proteinGroups.txt" table of MaxQuant. Proteins identified with $< 2$ unique peptides were removed from the analysis.

### Statistics and computational methods

In all experiments, data are presented as mean $\pm$ SEM unless stated otherwise. The number of biological replicates is defined in the figure legends. For flow cytometry experiments and cell proliferation analyses, statistical significances were calculated using two-sided paired Welch Student's *t*-test. t-SNE analysis was performed on means of intensities of phosphosites that were calculated between biological replicates and scaled by the maximal value observed over the course of TCR stimulation. Determination of the clusters was performed with the ClusterX function with default parameters from the R package ClusterX (https://github.com/JinmiaoChenLab/ClusterX) based on Rodriguez *et al*'s algorithm (Rodriguez & Laio, 2014). Within the t-SNE plot, *P*-value corresponding to the local enrichment of an annotation term (UniProt keyword or kinase–substrate relationship) in the neighborhood of an annotated site was calculated by comparing, using an hypergeometric test, annotations between the subset of sites that fell within a maximum distance of 4 (Euclidean distance calculated with t-SNE coordinates) and the set of all TCR-regulated sites. The databases of the April 2019 release of UniProt and the January 2019 version of Phosphosite were downloaded and used for the analyses.

### Western blot, antibodies

For biochemistry analysis, CD4$^+$ T cells were resuspended in a volume of 50 µl and stimulated for indicated times at 37°C with 50 µl biotinylated anti-CD3 (145-2C11) (3 µg), biotinylated anti-CD4 (GK1.5) (3 µg), and avidin (6 µg). Control unstimulated cells were incubated with avidin alone. Stimulation was stopped by the addition of 100 µl of a twice concentrated lysis buffer (100 mM Tris, pH 7.5, 270 mM NaCl, 1 mM EDTA, 20% glycerol, 0.4% n-dodecyl-β-maltoside) supplemented with protease and phosphatase inhibitors. After 10 min of incubation on ice, cell lysates were centrifuged at 21,000 *g* for 5 min at 4°C. Post-nuclear lysates were used for whole-cell lysates for subsequent immunoblot analysis. To obtain cytoplasmic and nuclear fractions, cells were lysed in buffer containing 25 mM Tris, 0.5 mM EDTA, and 0.25% Nonidet P-40 supplemented with protease and phosphatase inhibitors, and centrifuged at 4°C for 5 min at 14,000 rpm. Cytosolic fractions were collected, and nuclear fractions were suspended in nuclear lysis buffer (20 mM HEPES, 0.4 M NaCl, 1 mM EDTA, 1 mM EGTA, 1 mM DTT, 1% Triton X-100). The following antibodies were used for immunoblot analysis: anti-CBL-pY774 (3555), anti-ZAP-pY319/352 (2701), anti-AKT-pT308 (2965), anti-PLGg1-pY783 (2821), anti-ERK1/2-pY204/T202 (9106), anti-ERK1/2 (9102), anti-VAV1 (2502), anti-FOXO3-pS252 (5538), anti-FOXO3 (12829), anti-NFATC2 (5862), anti-RPS6-pS235/236 (4858), anti-P70S6K-pT389 (9206), anti-ACACA-pS79 (3661), anti-ACACA (3676), anti-TBCDC1 (4629), anti-AMPKa1−pT172 (2535), and anti-AMPKa1 (2795) from Cell Signaling Technology; anti-LCK (sc-433) from Santa Cruz Biotechnology; anti-PDCD4-pS457 (PA5-38806) from Thermo Fisher; and global anti-pY (4G10) and anti-TBCDC1-pS231 (07-2268) were purchased from Millipore. The anti-ITSN2 was kindly provided by A Rynditch (Novokhatska *et al*, 2013).

### Mass cytometry analysis

For short-term stimulation, Cas9-EGFP OT-I CD8$^+$ T cells transfected with sgEGFP were stained with 1 µM CFSE for 10 min at 37°C. $2 \times 10^6$ CFSE-stained, sgEGFP-transfected cells were mixed with an equal number of sgITSN2-1- or sgITSN2-2-transfected cells. Mixed cells were then incubated with 1 µM cisplatin (Cell-ID Cisplatin from Fluidigm, CA, USA) to enable subsequent identification of dead cells. Cells were then left unstimulated or stimulated with specified doses of N4 peptide MHC tetramers for the indicated times before proceeding to mass cytometry analysis. For long-term stimulation, Cas9-EGFP OT-I CD8$^+$-transfected T cells were harvested, washed, stained with 1 µM of cisplatin (for subsequent identification of dead cells), and stimulated for 6 h with indicated concentration of N4 peptides or PMA/ionomycin.

### Barcoding, labeling, and acquisition

To stop stimulation, cells were fixed with 2% PFA for 15 min at room temperature. Fixed cells were washed and barcoded using palladium isotopes (Cell-ID™ 20-Plex Pd Barcoding Kit from Fluidigm, CA, USA). $2.5 \times 10^6$ barcoded cells were then stained with metal-conjugated antibodies directed against surface antigens, washed, and permeabilized with cold methanol (95%) for 20 min. After permeabilization, cells were washed and stained with metal-conjugated antibodies directed against intracellular antigens before acquiring data on a Helios mass cytometer (Fluidigm, CA, USA). Antibodies used for surface and intracellular staining are listed in Appendix Table S1.

## Data availability

The mass spectrometry proteomic data have been deposited to the ProteomeXchange Consortium (http://proteomecentral.proteomexchange.org) via the PRIDE partner repository with the data set identifiers PXD014225 (CD4$^+$ T cells stimulated with anti-CD3/anti-CD4 antibodies) and PXD016583 (OT-I CD8$^+$ T cells stimulated with MHC-I tetramers loaded with N4 agonist peptide).

**Expanded View** for this article is available online.

## Acknowledgements

We thank N. Jarmuzynski, E. Maturin, and S. Durand for their technical help, as well as Y. Couté, C. Schaeffer, AM. Hesse, C. Ramus, A. Hovasse, S. Ferré, and E. Same for their assistance in the optimization of phosphoproteomic methods. This work was supported by CNRS, INSERM, the Agence Nationale de la Recherche (LymphoScan project to R.R), the European Research Council (ERC) under FP7 Program (grant agreement no. 322465 (INTEGRATE) to B.M.), and the European Union's Horizon 2020 Research and Innovation Program (grant agreement no. 787300 (BASILIC), to B.M.), the MSDAVENIR Fund (to B.M.), the Investissement d'Avenir program of the French Ministry of Research ProFI (Proteomics French Infrastructure, ANR-10-INBS-08 to O.B.-S. and J.G.) and PHENOMIN (French National Infrastructure for mouse Phenogenomics; ANR10-INBS-07 to B.M.); and fellowships from MSDAVENIR (Y.O. and D.M.), ERC INTEGRATE (M.G.N.), and PHENOMIN (L.G.) projects.

## Author contributions

RR, AGP, and GV conceived the project; ML-P, CF, MGM, and YO performed the experiments with the help of LG; CG and DM set up the experimental procedure to inactivate genes using the CRISPR/Cas9 system in primary T cells; CF, ML-P, and AGP performed the MS experiments and analysis; MMar and HL performed the mass cytometry analysis; GV and ML-P designed the computational and bioinformatic analysis; OB-S, JG, MMal, and BM provided key insights; and GV, ML-P, AGP, and RR wrote the manuscript with feedback from BM.

## Conflict of interest

The authors declare that they have no conflict of interest.

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
