## [Review Process File · Molecular Systems Biology]

LymphoAtlas: a dynamic and integrated phosphoproteomic resource of TCR signaling in primary T cells

Marie Locard-Paulet, Guillaume Voisinne, Carine Froment, Marisa Goncalves Menoita, Youcef Ounoughene, Laura Girard, Claude Gregoire, Daiki Mori, Manuel Martinez, Hervé Luche, Jérôme Garin, Marie Malissen, Odile Schiltz, Bernard Malissen, Anne Gonzalez de Peredo, and Romain Roncagalli

DOI: 10.15252/msb.20209524

Corresponding author(s): Romain Roncagalli (roncagalli@ciml.univ-mrs.fr), Anne Gonzalez de Peredo (gonzalez@ipbs.fr)

Review Timeline:

Submission Date:	17th Feb 20
Editorial Decision:	26th Mar 20
Revision Received:	9th Apr 20
Editorial Decision:	15th May 20
Revision Received:	19th May 20
Accepted:	20th May 20

Editor: Jingyi Hou

Transaction Report:

26th Mar 2020

Manuscript Number: MSB-20-9524

Title: LymphoAtlas: a dynamic and integrated phosphoproteomic resource of TCR signaling in primary T cells

Author: Marie Locard-Paulet

Guillaume Voisinne

Carine Froment

Marisa Goncalves Menoita

Youcef Ounoughene

Laura GIRARD

Claude Gregoire

Daiki Mori

Manuel Martinez

Hervé Luche

Jérôme Garin

Marie Malissen

Odile Schiltz

Bernard Malissen

Anne Gonzalez de Peredo

Romain Roncagalli

Dear Dr Roncagalli,

Thank you again for submitting your work to Molecular Systems Biology. We have now heard back from the three reviewers who agreed to evaluate your study. As you will see below, the reviewers think that the study is interesting and they acknowledge the quality and potential relevance of the presented data. However, they raise a series of concerns, which should be carefully addressed in a revision of the manuscript. Since the reviewers' recommendations are rather clear, there is no need for me to reiterate all the points listed below.

On a more editorial level, please do the following:

- Please provide a .docx formatted version of the manuscript text (including legends for main figures, EV figures and tables). Please make sure that the changes are highlighted to be clearly visible.

- Please provide individual production quality figure files as .eps, .tif, .jpg (one file per figure).

- Please provide a .docx formatted letter INCLUDING the reviewers' reports and your detailed point-by-point responses to their comments. As part of the EMBO Press transparent editorial process, the point-by-point response is part of the Review Process File (RPF), which will be published alongside your paper.

- Please note that all corresponding authors are required to supply an ORCID ID for their name upon submission of a revised manuscript.

- Since we only allow a maximum of 5 EV Figures, additional EV figures should be bundled together with their legends in a single PDF file called *Appendix*, which should start with a short Table of Content. Appendix figures (and Tables) should be referred to in the main text as: "Appendix Figure S1, Appendix Figure S2...Appendix Table S1" etc. Each legend should be below the corresponding Figure/Table in the Appendix. See detailed instructions regarding expanded view here: <https://www.embopress.org/page/journal/17444292/authorguide#expandedview>.

- We would encourage you to include the source data for figure panels that show essential quantitative information. Additional information on source data and instruction on how to label the files are available at < <https://www.embopress.org/page/journal/17444292/authorguide#sourcedata> >.

- All Materials and Methods need to be described in the main text. We would encourage you to use 'Structured Methods', our new Materials and Methods format. According to this format, the Material and Methods section should include a Reagents and Tools Table (listing key reagents, experimental models, software and relevant equipment and including their sources and relevant identifiers) followed by a Methods and Protocols section in which we encourage the authors to describe their methods using a step-by-step protocol format with bullet points, to facilitate the adoption of the methodologies across labs. More information on how to adhere to this format as well as downloadable templates (.doc or .xls) for the Reagents and Tools Table can be found in our author guidelines: < <https://www.embopress.org/page/journal/17444292/authorguide#researcharticleguide> >. An example of a Method paper with Structured Methods can be found here: .

- Please provide a "standfirst text" summarizing the study in one or two sentences (approximately 250 characters, including space), three to four "bullet points" highlighting the main findings and a "synopsis image" (550px width and max 400px height, jpeg format) to highlight the paper on our homepage.

- When you resubmit your manuscript, please download our CHECKLIST (http://embopress.org/sites/default/files/Resources/EP_Author_Checklist.xls) and include the completed form in your submission. *Please note* that the Author Checklist will be published alongside the paper as part of the transparent process <http://msb.embopress.org/authorguide#transparentprocess>.

If you feel you can satisfactorily deal with these points and those listed by the reviewers, you may wish to submit a revised version of your manuscript. Please attach a covering letter giving details of the way in which you have handled each of the points raised by the referees. A revised manuscript will be once again subject to review and you probably understand that we can give you no guarantee at this stage that the eventual outcome will be favorable.

I look forward to receiving your revised manuscript soon.

Yours sincerely,

Jingyi Hou
Editor
Molecular Systems Biology

If you do choose to resubmit, please click on the link below to submit the revision online *within 90 days*.

Link Not Available

IMPORTANT: When you send your revision, we will require the following items:

1. the manuscript text in LaTeX, RTF or MS Word format
2. a letter with a detailed description of the changes made in response to the referees. Please specify clearly the exact places in the text (pages and paragraphs) where each change has been made in response to each specific comment given
3. three to four 'bullet points' highlighting the main findings of your study
4. a short 'blurb' text summarizing in two sentences the study (max. 250 characters)
5. a 'thumbnail image' (550px width and max 400px height, Illustrator, PowerPoint or jpeg format), which can be used as 'visual title' for the synopsis section of your paper.
6. Please include an author contributions statement after the Acknowledgements section (see <https://www.embopress.org/page/journal/17444292/authorguide>)
7. Please complete the CHECKLIST available at (<http://bit.ly/EMBOPressAuthorChecklist>). Please note that the Author Checklist will be published alongside the paper as part of the transparent process (<https://www.embopress.org/page/journal/17444292/authorguide#transparentprocess>).
8. Please note that corresponding authors are required to supply an ORCID ID for their name upon submission of a revised manuscript (EMBO Press signed a joint statement to encourage ORCID adoption). (<https://www.embopress.org/page/journal/17444292/authorguide#editorialprocess>)

Currently, our records indicate that there is no ORCID associated with your account.

Please click the link below to provide an ORCID:

Link Not Available

The system will prompt you to fill in your funding and payment information. This will allow Wiley to send you a quote for the article processing charge (APC) in case of acceptance. This quote takes into account any reduction or fee waivers that you may be eligible for. Authors do not need to pay any fees before their manuscript is accepted and transferred to the publisher.

*** PLEASE NOTE *** As part of the EMBO Press transparent editorial process initiative (see our Editorial at <http://dx.doi.org/10.1038/msb.2010.72>), Molecular Systems Biology publishes online a Review Process File with each accepted manuscripts. This file will be published in conjunction with your paper and will include the anonymous referee reports, your point-by-point response and all pertinent correspondence relating to the manuscript. If you do NOT want this File to be published, please inform the editorial office at msb@embo.org within 14 days upon receipt of the present letter.

Reviewer #1:

The manuscript by Locard-Paulet et al. describes the phosphorylation landscape of primary t-cells upon stimulation of the TCR, at very early timepoints (15 and 30 sec.) up to 600 seconds. The authors enrich both pS/pT and pY sites and identify a large number of sites active in different processes of t-cell activation and function. The authors perform elaborate data analysis and describe several processes and regulatory observations highlighting the importance of their dataset. Finally, the authors validate ITSN2 as a scaffolding protein important in regulating TCR surface expression. Overall, the amount of work performed is impressive, the manuscript is well written, and the authors have managed to highlight the informative potential of the dataset, which indeed can serve as a valuable resource and fits well in MSB.

A few comments on the manuscript below.

- How are cells stimulated with anti-CD3 and anti-CD4 antibodies (in a homogenous fashion), followed by streptavidin crosslinking and harvesting of cells within the mentioned timepoints of 15 and 30 seconds. What is the reproducibility of these early timepoints? Where cells washed before snap freezing?
- The authors choose relatively loose cutoff criteria for determining 'regulated' sites. There are however, several sites with large p-values ($-\log_{10}$ of 8-10) and/or fold change (~ 4). It would be interesting to know which sites these are.
- Figure 2C, color differences are very hard to differentiate, especially the 3 greens and last 2 blues. It seems the tyrosine phosphorylations on the receptor are located in cluster 7 (or 8?), while here one would expect a rapid increase as observed in clusters 1-4. Can the authors comment on this observation? Also, how is the rapid increase of sites in cluster 1-4 downstream of the receptor explained, when the receptor itself responds much slower?
- In line with this, is the observation of the striking decrease of phosphorylation in, especially, clusters 9-10 at 15sec. followed by a marked increase at 30sec. (This later comes back in EV3b with the phosphorylation of CBX4-S350). Is this real or could it be an effect of imputation, or related to variation in this very early timepoint? I would advise to check these phosphorylations manually, since these look odd (artificial).
- In figure legend 3a and 4a, "(hypergeometric test P-value {less than or equal to} 0.05, fold-change {greater than or equal to} 2, number of annotated sites {greater than or equal to} 2; Grey squares: fold-change < 1)" Where the author state the fold-change {greater than or equal to} 2, and grey squares: fold-change < 1, but the color bar indicate from 0.0 to 2.0. To my knowledge, $\text{Log}_2(2) = 1$.

Minor:

- The authors sometimes use 'big words' for their workflow and results, while these are actually rather standard in the current proteomics community and can be toned down a bit (notwithstanding the impressive amount of work included in the manuscript). Some examples;
 - introduction: "Here, we have developed such an MS-based approach to.." what is exactly the new development?
 - Abstract "Using an integrated and unsupervised analysis", also standard for proteomics/phosphoproteomics.
 - "deep proteome generated using SDS-PAGE pre- fractionation", the 'deep proteome' is simply the proteomics part of the analysis.

- The authors describe a "Stringent workflow" for quantification, however the chosen cutoffs are not very stringent (relatively small fold change (1,75) and low p-value (0,05)) and are arbitrarily chosen? which can be accepted but mention as arbitrary cutoff based on distribution.

- repetition of visualization of phosphorylated amino acids in figure 1b and 1c are the same as in EV1b.
- Use of past tense in abstract and introduction
- How do cell surface receptors scan?
- In the results the authors report 6,684 phosphosites and in the discussion 7,180 as well as 254 vs 349 pY sites. Is this the result of combining 2 datasets in the discussion? Make this clear.

Reviewer #2:

The manuscript presented by Locard-Paulet et al. presents an important resource for TCR signaling in primary T cells. The experimental procedures and data analysis are of high quality and the paper is well and logically written. Primary CD4⁺ T cells were first briefly expanded and then stimulated by anti-CD3 and anti-CD4 ligation. Experiments were done in biological replicates from independent mice (n=4) + technical replicates to increase coverage of phosphosites (n=3). The experimental design is well balanced and the authors produced a deep dataset including a large number of usually hard to cover phospho-tyrosines. In addition, the data is available in a dedicated webtool. The time-series data has high temporal resolution and signaling events are presented using an elegant data visualization approach. Importantly, the results in CD4⁺ T cells stimulated by anti-CD3/CD4 are largely validated by a second experiment using stimulation of primary OT-I CD8⁺ T cells that express a TCR specific for the ovalbumin (OVA)-derived SIINFEKL (N4) peptide bound to H-2Kb MHC-I molecules. Finally, the authors characterize a novel role of the phosphoprotein ITSN2 as a regulator of TCR signaling and receptor internalization.

The reviewer has no major concerns about this nice paper, which has clear relevance and impact for the field. Below some suggestions for improvement.

Comments:

- It would have been interesting to better integrate the pSTY data presented in this manuscript with the authors recently published TCR signaling interactomes from CD4 T cells (<https://www.ncbi.nlm.nih.gov/pubmed/31591574>). If the authors could analyze the dynamics of their phosphosites in respect to their presence in distinct protein complexes they may come up with a refined model and predictions of how these sites could regulate the dynamics of protein complex formation in TCR signaling. Can the authors do this analysis at least for the discussed role of ITSN2 in TCR internalization?
- The correlation of phosphosite occupancy with nuclear/cytoplasm ratio of the protein is validated for NFATC2 and FOXO3. It would be better to approach this important functional aspect of the reported phosphosites globally and evaluate nuclear/cytoplasm ratios after stimulation not by WB as shown in Fig EV3, but rather by a proteomic approach. Can the authors perform mass spec on samples prepared like in Fig EV4c and correlate the site occupancy in their pSTY datasets with nuclear/cytoplasm ratios for all proteins? This would be a valuable addition to the dataset.

Minor comments:

- Figure EV1c: labelling of y axis is missing
- Numbers in the text of the unique phosphorylation sites don't correspond to the numbers in Figure EV1b
- Figure EV2: figure captions for EV2a and EV2b are mixed-up
- Figure 4e: total ERK1/2 is missing in the WB

Reviewer #3:

In this manuscript, Locard-Paulet et al provide a rigorous analysis of the phosphoproteome of CD4 T cells stimulated by antibody or MHC tetramer through multiple timepoints. Although others have attempted to characterize the primary T cell phosphoproteome previously, those studies lacked the rigor and in-depth computational analysis to truly exploit this type of data. This study is unique both in the number of timepoints examined as well as the excellent follow-up study on the intersectin 2 protein that was uncovered. Showing that wide-scale analysis uncovered the modulation of the T cell activation threshold by ITSN2 was an excellent proof of concept that this data has great utility. Overall, this study provides the most rigorous view to date of the T cell phosphoproteome in primary T cells and provides modes of analysis of the large wealth of quantitative phosphorylation information that not only reveal new signaling pathways in T cells but set the bar for future studies beyond T cells to extract biological meaning out of these sorts of wide-scale experiments. This is a fantastic window into T cell signaling.

Major Questions:

1. Primary T cell phosphoproteomics has been performed previously and the methods employed to make the phosphoproteomic measurements in this study are not intended to be novel. The introduction in particular should compare and contrast the current work with the previous efforts. In particular, the 2017 Front. Immunol paper by Joshi et al and the 2012 article in MCP by Ruperez et al and the 2014 article by Navarro et al in MCP should be compared and contrasted with the current study. The Joshi study in particular did compare naive to CD3/28 stimulated primary human T cells and included a rigorous number of replicates and a large number of sites identified. The Ruperez et al study also looked at both the total and pY phosphoproteome, very similarly to this study. I think there is a very strong argument that this study transcends these others but I feel like it needs to be made more explicitly to highlight the novelty. The discussion mentions that this manuscript focuses "for the first time" on early phosphorylation events but the Joshi et al paper published in 2017 did compare 5 minute CD3/28 stimulated to naive human CD4 T cells. The authors also mention in the introduction "despite these efforts, a global understanding of this complex signaling network is still lacking". Could you please be more specific on how the other primary T cell phosphoproteomics studies have fallen short to showcase the novelty of this study. I would say that the computational analysis of your study transcends anything done previously and transforms the study into a more generally useful resource. Also consider rewording "we have developed" on page 4 to indicate that you are using biochemical methods that others have developed i.e. TiO2/ pTyr100 IP and their combination (see Ruperez et al and numerous other papers that have used the same methods). Certainly, this is not a method development paper where phosphoproteomic methods are being developed and optimized.

2. Unlike most other studies, this study also runs the control experiment to test whether protein abundance has changed significantly to explain a subset of the phosphorylation changes. The statement is made on page 6 that "protein abundances were not impacted by TCR activation at

these early timepoints" but surely only some fraction of the proteins with observed phosphorylation sites were detected in the total proteome analysis as only 6500 proteins were quantified according to figure EV2. Looks like maybe 10-20% were not quantified according to EV2. Could you please report this percentage in the text of the manuscript and indicate that this information for individual proteins is available in your web tool. The concern is that a T cell signaling researcher might be interested in a particular site and not know to check for the unphosphorylated protein data or to collect western blot data on this protein if the particular protein was not observed. For example, for the sites featured in Figure 2c, could you mark differently the sites on proteins that were not captured in the unphopsho dataset or if they were all captured, state that?

3. One of the most interesting parts of your data was the relatively slow kinetics of ITAM CD3 subunit and zeta phosphorylation relative to phosphorylation sites typically believed to be downstream of ITAM phosphorylation. For example, could you please discuss why Erk, PLCg1, Lck, and Zap70 phosphorylation appear to happen much faster than ITAM phosphorylation. Do you suspect that ITAMs are not leading to phosphorylation of these sites in your cells? Also, the way the data is presented as cluster patterns, it is hard to tell which timepoints are significantly changed. I assume that the requirement for inclusion in this figure is that at least one but not necessarily all timepoints must be significantly altered. A supplemental heatmap figure for data in fig 2c showing exactly which timepoints are significant relative to the 0 min point would be helpful.

Minor Questions:

1. page 3, not sure "signalization cascades" is a word.

2. I was a bit confused when you mentioned at the bottom of page 10 the phosphorylation of putative Erk and Akt sites on STMN1 and other proteins. Are these predictions of the direct phosphorylation of these proteins in general by ERK/Akt in phosphosite or sites that show an Erk/Akt motif. Are these Erk/Akt pathway or direct targets? Are the interactions direct or indirect? Maybe you could explain in a bit more detail the phosphosite kinase-substrate relationship data that you are using here.

3. Not really a question but a comment that the degree of overlap of the OT-I and anti-CD3/4 dataset was really a strong supporter of the quality of your data. In particular supplemental figure EV4b-d and especially EV4d was very powerful and you may consider upgrading it into the regular figures from the supplement. I am not aware that tetramer stimulation phosphoproteomic analysis has been performed previously in primary T cells and you might want to hype this up more in your intro.

4. For Figure EV4b, consider switching to a pseudocolor dot figure to indicate the density instead of contour plot although this might just be a point of preference.

Point-by-point reply

General comments for the three Reviewers

We are glad to see that the three Reviewers appreciated our study. The present point-by-point reply addresses the important issues that were raised and we provide below an outline of the requested rectifications that have been incorporated in our revised manuscript.

Reviewer #1:

The manuscript by Locard-Paulet et al. describes the phosphorylation landscape of primary t-cells upon stimulation of the TCR, at very early timepoints (15 and 30 sec.) up to 600 seconds. The authors enrich both pS/pT and pY sites and identify a large number of sites active in different processes of t-cell activation and function. The authors perform elaborate data analysis and describe several, processes and regulatory observations highlighting the importance of their dataset. Finally, the authors validate ITSN2 as a scaffolding protein important in regulating TCR surface expression. Overall, the amount of work performed is impressive, the manuscript is well written, and the authors have managed to highlight the informative potential of the dataset, which indeed can serve as a valuable resource and fits well in MSB.

We thank Reviewer #1 for summarizing and highlighting the strengths of our work.

A few comments on the manuscript below.

- How are cells stimulated with anti-CD3 and anti-CD4 antibodies (in a homogenous fashion), followed by streptavidin crosslinking and harvesting of cells within the mentioned timepoints of 15 and 30 seconds. What is the reproducibility of these early timepoints? Where cells washed before snap freezing?

We thank Reviewer #1 for the opportunity to clarify our protocol of stimulation. As indicated in Materials and Methods purified T cells were washed in serum free media, co-incubated with anti-CD3 and anti-CD4 biotinylated antibodies at 37°C and then cross-linked with streptavidin for the desired time. To tightly control the time of stimulation, cells were immediately snap frozen in liquid nitrogen (with no wash). This stopped the stimulation and fixed the molecular state associated with each time point (including those of 15 and 30 seconds). Regarding the reproducibility of the early time points, the distribution of the coefficients of variation (CV= standard deviation divided by the mean) for each phosphorylation site across all biological replicates shows very similar results for all time points and are of weak amplitude, indicating a high reproducibility of all conditions of stimulation (see below, medians are indicated for each condition/time point).

- The authors choose relatively loose cutoff criteria for determining 'regulated' sites. There are however, several sites with large p-values ($-\log_{10}$ of 8-10) and/or fold change (~ 4). It would be interesting to know which sites these are.

We thank Reviewer #1 for his/her comment. Although we follow a clear statistical procedure, we admit that the cutoff criteria chosen (Tukey corrected p-value $\leq 0,05$ and fold change $\geq 1,75$) are arbitrary, as noted below by reviewer, and that it can be interesting to provide a shortlist of very confident and strongly regulated phosphorylation sites. We thus filtered the data using the very stringent threshold values suggested by Reviewer #1. We identified 18 sites following these criteria (Tukey pval $< 10^{-8}$ and fold change > 4 -see the list below, for sake of space the Anova p-values are displayed). This list includes CD3e and several components involved in the cytoskeleton reorganization such as members of the beta-thymosin family (Tmsb4 and Tmsb10) regulating actin polymerization, Stamin 1 (Stmn1) controlling microtubule formation and Spectrin alpha chain, non-erythrocytic 1 (Sptan1) connecting elements of the cytoskeleton.

GeneID	pAnova	BestFC (log2)	Accession
Tmsb4x-T29	7,0025E-17	5,61294086	P20065
Rin3-S389	7,2672E-13	4,14013392	P59729
Tmsb10-T23	3,3648E-11	5,27496377	Q6ZWY8
Cd3e-Y181	2,1618E-09	6,26341476	P22646
Stmn1-S25	1,0379E-10	4,05677977	P54227
Nop56-S554	3,8624E-13	2,60188865	Q9D6Z1
Grap2-T254	7,8158E-14	2,38465303	O89100
Fancd2-S10	2,8556E-09	3,37202676	Q80V62
Lcp1-S5	1,3245E-10	3,92608192	Q61233
Samsn1-S90	2,9119E-09	3,1094651	P57725
Rcsd1-S177	5,9463E-10	3,097335	Q3UZA1
Ppp1r9b-S100	5,2628E-10	3,54834844	Q6R891
Sptan1-S1031	5,1776E-11	6,22893883	P16546

Dbnl-S277	1,4046E-10	3,70090999	Q62418
Stmn1-			
S16+S25	2,4442E-11	3,69922773	P54227
Sp110-S175	3,0484E-09	2,39925294	Q8BVK9
Arhgef1-S905	2,6825E-10	2,09875623	Q61210
Tpr-S2223	1,0293E-08	4,30637944	F6ZDS4

However, using such low p-value threshold, we eliminated many sites which are truly regulated in the system, as known from the literature: for examples sites from ERK1 and ERK2 (Mapk1-Y185 and Mapk3-Y205), which represent important effectors of the TCR pathway, but were measured with p-values around 10^{-5} . As a trade-off, we selected only the sites with a fold change > 6 and Tukey corrected p-value $< 10^{-5}$. These threshold values yielded a list of 63 sites, that we feel is representative of confident and very strongly regulated sites downstream of the TCR, and still small enough to generate a readable table that can be useful for the reader. We have incorporated this list in Dataset EV1. This is now mentioned in the “Result section” of the modified version of the manuscript (page 8):

“Noticeably, among the 730 phosphorylation sites (or combination of sites) that we identified as significantly regulated according to the criteria described above, some exhibited very strong regulation, with fold changes of high amplitude (Dataset EV1).”

GeneID	NS	15	30	120	300	600	pAnova	BestFC	Cluster
Wipf1-S330							2,97162E-10	2,7420748	1
Pak2-S197							1,70548E-08	3,758068378	1
Rasal3-S883							5,66995E-07	2,870740383	1
Tmsb10-T23							3,36481E-11	5,274963769	2
Tmsb4x-T29							7,00253E-17	5,612940857	2
Sash3-S97							5,90501E-08	5,159357254	2
Ppp1r9b-S100							5,26279E-10	3,548348436	2
Dbnl-S277							1,40462E-10	3,70090999	2
Cbl-Y672							9,46801E-07	5,649584996	2
Dock11-S12							5,79482E-09	3,300280146	2
Mapk1-Y185							4,05452E-06	4,798902267	2
Map4k1-S325							2,98924E-08	3,249071426	2
Pacs1-S526							2,54531E-09	3,867032448	3
Ubash3a-Y9							3,00951E-08	5,325830566	3
Zap70-Y290							5,56628E-08	6,687740952	3
Zap70-Y491							4,52595E-08	5,294417818	3
Fam65b-S582							1,7924E-07	3,931046216	3
Fyb-S203							5,26038E-07	4,62498945	3
Lsp1-S184							5,88528E-06	4,174051247	3
Cbl-S617							1,62639E-08	2,663376883	3
Zap70-Y492							2,49676E-08	4,618431965	4
Rin3-S389							7,26725E-13	4,140133925	4
Samsn1-S90							2,91188E-09	3,109465097	4
Grap2-S230							1,51374E-08	2,904912597	4
Dock8-S433							1,13319E-08	2,654829728	4
Ppp1r12a-S507							1,90377E-08	3,727800986	5
Nelfe-S51							4,08701E-08	3,3495356	5
Rcsd1-S127							2,1496E-06	4,715874203	5
Mapk3-Y205							1,10334E-05	6,245479129	5
Ncoa2-S771							6,09334E-07	2,794210514	5
Cd3e-Y170							6,11224E-07	5,514498315	6
Tpr-S2223							1,02933E-08	4,306379437	6
Asap1-S1059							1,39691E-07	3,643883838	6
Utp14a-S567							3,36907E-07	3,166379779	6
Fancd2-S10							2,85556E-09	3,372026763	6
Rasal3-S95							8,2751E-08	2,802105644	6
Stmn1-S16+S25							2,44421E-11	3,699227727	6
Trim28-S489							5,02143E-07	3,537670451	6
Rcsd1-S108							9,06921E-06	3,397588734	6
Sun2-S12							6,40957E-06	3,607034152	6
Pola1-S192							1,36051E-09	2,909987703	6
Reps1-S708							6,72131E-07	2,6353199	6
Ncl-S577							1,78E-06	2,651534534	6
Cd3e-Y181							2,16182E-09	6,263414759	7
Serbp1-S329							1,00178E-08	4,422699517	7
Cd247-Y111/111							4,3867E-08	5,07558906	7
Cd247-Y123/123							4,58448E-08	4,768744816	7
Lcp1-S5							1,32454E-10	3,92608192	7
Ptpn7-S143							9,19071E-08	3,261849584	7
Map4k1-S405							7,20286E-09	3,512705412	7
Stmn1-S25							1,03788E-10	4,056779765	7
Ncbp1-S22							1,42259E-07	4,364507925	7
Rcsd1-S177							5,94631E-10	3,097335002	7
Rgs10-S16							2,70163E-08	3,243764188	7
Cd247-Y142							9,23727E-06	4,69802298	7
Rcsd1-T243+S246							8,96103E-07	4,412644538	8
Whsc1l1-S457							1,1694E-08	3,041021152	8
Nop56-S554							3,86238E-13	2,601888647	8
Whsc1l1-T456							9,22483E-07	2,585331249	8
Zc3h14-S309							7,60679E-07	2,614896918	8
Sptan1-S1031							5,17761E-11	6,228938832	12
Sptan1-S1029							2,20997E-07	3,40677472	12
Ddx5-S24							8,0523E-07	2,94020265	13

- Figure 2C, color differences are very hard to differentiate, especially the 3 greens and last 2 blues. It seems the tyrosine phosphorylations on the receptor are located in cluster 7 (or 8?), while here one would expect a rapid increase as observed in clusters 1-4. Can the authors comment on this observation? Also, how is the rapid increase of sites in cluster 1-4 downstream of the receptor explained, when the receptor itself responds much slower?

We thank Reviewer #1 for his/her constructive comment.

The colors have been modified to improve clarity in the modified version of the manuscript.

As interestingly noticed by Reviewer #1, cluster 7 is enriched in receptors. Particularly it contains many phosphorylation sites from the CD3 chains. We understand that based on the unique cluster numeration, one could conclude that receptor phosphorylation occurs after phosphorylation of the proteins localized in C1-4. This is actually not the case. In fact, the clustering is really based on the shape of the kinetics, it is performed after scaling of the data and thus, it doesn't take the amplitude into account. The induction of the phosphorylation of CD3 chains is actually very rapid and is already intense at 15 seconds but not yet maximum (see figure below). This phosphorylation increases progressively during the stimulation and reaches a maximum between 300 and 600 seconds. In contrast, the phosphorylation of proteins in C1-C4 reach their maximum very rapidly (typically at 15 or 30s) and some are very transient. For example, phosphorylation at position 185 of Mapk1, a key effector downstream of the receptor, is maximal at 30s and is downregulated afterwards (cluster 2). As illustrated below, the overlay of phosphorylation dynamics of CD3-Y142 and MAPK1-Y185 shows that their inductions and amplitudes are similar (the intensity has been normalized on the NS condition).

We wrote in the text that clusters C5-C8 had a “slower dynamic”, which is maybe misleading. In the revised version, we qualified the phosphorylation profiles as “transient” (clusters 1-2), “sustained” (3-6), or “progressive” (7-8).

The description of clusters 7 and 8 is now as follows (page 9):

“For C7 and C8, we observed a gradual and continuous amplification of the phosphorylation signal all along the time-course, which remained sustained (C7) or was even increasing (C8) at the later time points”

Regarding the progressive phosphorylation of the TCR, one explanation could be that the phosphorylation of only a few CD3 chains is sufficient to trigger a large wave of phosphorylation of downstream proteins (C1-C4) (Huang, J. et al. *Immunity* 39, 846–857, 2013). The subsequent and progressive phosphorylation increase of the CD3 chains could be the result of the positive feedback mechanism involving the stabilization of the active conformation of Lck promoting a progressive increase of the ZAP70 kinase activity combined with the protection of phosphorylated ITAMs from surrounding phosphatases by the binding of ZAP70. In this scenario, ITAM phosphorylation could stay high even though the phosphorylation of downstream effectors drops. In a non-exclusive manner, this phenomenon could also reflect a progressive accumulation of endocytosed CD3 chains that remain phosphorylated intracellularly.

As mentioned in the discussion of the manuscript, this observation remains fully compatible with known models of TCR activation.

- In line with this, is the observation of the striking decrease of phosphorylation in, especially, clusters 9-10 at 15sec. followed by a marked increase at 30sec. (This later comes back in EV3b with the phosphorylation of CBX4-S350). Is this real or could it be an effect of imputation, or related to variation in this very early timepoint? I would advise to check these phosphorylations manually, since these look odd (artificial).

We thank Reviewer #1 for noticing this point. As Reviewer #1 mentioned, clusters 9-10 are characterized by a quick and transient decrease of phosphorylation. When checked manually, and for some of them, intensity values have been imputed for time points where the peptide was not detected (see below, triangles indicate when values have been imputed). However, imputation follows strict rules in our analysis:

- if for a given condition (time point), a phosphorylation site was not quantified in any replicate experiment or quantified in only one of the replicates, missing values for that condition were imputed with a low-intensity value, obtained from a draw around the 5% quantile, and the standard deviation of the entire data set.
- To avoid biases coming from the random draw of replacement values, this statistical process was performed 200 independent times and phospho-sites were considered statistically significant if they presented a p-value of the ANOVA ≤ 0.05 and a minimum of one couple of time points with more than 2 points per condition and a Tukey corrected p-value ≤ 0.05 with an associated absolute fold change > 1.75 in a minimum of 90% of the iterations.

Also, we would like to remind that only phosphorylation sites that were detected in at least 3 conditions (time points), and presenting in each of them a minimum of two quantification values from independent replicate experiments were retained for statistical analysis.

With this pipeline (and like in most similar high throughput analysis), we fully agree that some of the intensity values were imputed to perform statistical analysis. This is the case for CBX4-S350 (see below). However, it is unlikely that these results are artificial since all independent biological replicates show an identical result for the considered time point, while values are present for other time points.

Moreover, such slight decrease can also be observed with phosphorylation sites (of the same clusters 9-10) for which real values have been measured in all biological replicates (see below).

Therefore, although it is related to a small number of phosphorylation sites (clusters 9 and 10 are rather small), these results suggest that such regulation is possible upon TCR stimulation.

That being said, we agree that fold changes calculated based on imputed values should be interpreted with caution. We also admit that for some of the sites of these clusters, measured from

real values, this initial and very transient downregulation is of small amplitude. We cannot exclude false positive hits, which always exist, and this is the reason why we provide a web interface that allows readers to manually check the data (raw, normalized and imputed) of all their sites of interest before drawing biological conclusions.

- In figure legend 3a and 4a, "(hypergeometric test P-value {less than or equal to} 0.05, fold-change {greater than or equal to} 2, number of annotated sites {greater than or equal to} 2; Grey squares: fold-change < 1)" Where the author state the fold-change {greater than or equal to} 2, and grey squares: fold-change < 1, but the color bar indicate from 0.0 to 2.0. To my knowledge, $\text{Log}_2(2) = 1$.

We thank Reviewer #1 for his/her comment. We understand that the confusion comes from our use of a \log_2 transformed color scale. On these plots, we have chosen to display fold-changes ranging from 1 to 4. Since we use \log_2 transformed values, the color scale ranges from 0 to 2. To make it clear we have displayed fold-change instead of $\log_2(\text{fold-change})$ values on the color scale in a modified versions of Figures 3 and 4.

Minor:

- The authors sometimes use 'big words' for their workflow and results, while these are actually rather standard in the current proteomics community and can be toned down a bit (notwithstanding the impressive amount of work included in the manuscript).

We thank Reviewer #1 for advising us to tune down some of our statements in the manuscript. Related modifications have been done in a modified version of the manuscript.

Some examples;

- introduction: "Here, we have developed such an MS-based approach to.." what is exactly the new development?

The introduction has been modified, (also to describe in more details previous proteomic studies performed on primary T cells, as required by Reviewer #3), and this sentence is now "*Here, our aim was to apply such unbiased, large-scale MS-based methods...*" (page 4).

- Abstract "Using an integrated and unsupervised analysis", also standard for proteomics/phosphoproteomics.

Here, we wanted to highlight the efforts we put in the bioinformatics analysis of the dataset. Indeed, we believe this is a strength of this paper (and Reviewer #3 advised to insist a bit more on this point). We agree that our previous phrasing was awkward, and that proteomic data are somehow always analyzed in an unsupervised manner. So we reformulated this sentence as: "*Bioinformatic analysis of the data revealed.....*"

- "deep proteome generated using SDS-PAGE pre- fractionation", the 'deep proteome' is simply the proteomics part of the analysis.

That has been modified each time "deep" was used.

- The authors describe a "Stringent workflow" for quantification, however the chosen cutoffs are not very stringent (relatively small fold change (1,75) and low p-value (0,05)) and are arbitrarily chosen? which can be accepted but mention as arbitrary cutoff based on distribution.

We used the word "Stringent" workflow as compared to most of the analyses on primary T cells that use only one of the two criteria, whereas here we used a combination of thresholds based on the significance (p-values) and fold change to determine the regulated phosphosites. However, we understand the point of view of the reviewer and removed the word "stringent".

- repetition of visualization of phosphorylated amino acids in figure 1b and 1c are the same as in EV1b.

We thank Reviewer #1 for having noticed this point. Redundant pie charts from EV1b have been removed in a modified version of the manuscript.

- Use of past tense in abstract and introduction

We thank Reviewer #1 for advising us on improving the form of the manuscript. Related modifications have been done in a modified version of the manuscript.

- How do cell surface receptors scan?

We assume that Reviewer #1 thinks that the word 'scan' is not appropriate to define the function of receptors. This word has been removed in a modified version of the manuscript.

- In the results the authors report 6,684 phosphosites and in the discussion 7,180 as well as 254 vs 349 pY sites. Is this the result of combining 2 datasets in the discussion? Make this clear.

We thank Reviewer #1 for having noticed this point. 6,684 phosphosites (including 254 pY sites) correspond to individual phosphorylation sites while what we called later in the discussion 7,180 phosphosites (including 349 pY) correspond in fact to unique plus combination of sites quantified together on the same peptides.

This is now clarified on page 5 of the "Results" section of the manuscript as follows:

*"These led to the identification of 13,009 unique phosphorylated peptides corresponding to 9,702 and 560 phospho-sites –or combination of sites in the case of multiple phosphorylated peptides– in the TiO₂ and the pY-IP data set, respectively (**Figure EV1b**). From these peptides, we were able after filtering (PEP value > 0.01, localization score >75%, elimination of sites with high number of missing values, see Material and Methods) to relatively quantify 7,180 phosphorylation sites or combinations of sites (**Dataset EV1**). These correspond to 6,984 unique, non-redundant sites (**Figure 1b-c**), including 5659 phospho-serines (pS), 1,070 phospho-threonines (pT) and 254 pY present on 2,118 proteins in primary T cells".*

Reviewer #2:

The manuscript presented by Locard-Paulet et al. presents an important resource for TCR signaling in primary T cells. The experimental procedures and data analysis are of high quality and the paper is well and logically written. Primary CD4+ T cells were first briefly expanded and then stimulated by anti-CD3 and anti-CD4 ligation. Experiments were done in biological replicates from independent mice (n=4) + technical replicates to increase coverage of phosphosites (n=3). The experimental design is well balanced and the authors produced a deep dataset including a large number of usually hard to cover phospho-tyrosines. In addition, the data is available in a dedicated webtool. The time-series data has high temporal resolution and signaling events are presented using an elegant data visualization approach. Importantly, the results in CD4+ T cells stimulated by anti-CD3/CD4 are largely validated by a second experiment using stimulation of primary OT-I CD8+ T cells that express a TCR specific for the ovalbumin (OVA)-derived SIINFEKL (N4) peptide bound to H-2Kb MHC-I molecules. Finally, the authors characterize a novel role of the phosphoprotein ITSN2 as a regulator of TCR signaling and receptor internalization.

The reviewer has no major concerns about this nice paper, which has clear relevance and impact for the field. Below some suggestions for improvement.

We thank Reviewer #2 for summarizing and highlighting the strengths of our work.

Comments:

- It would have been interesting to better integrate the pSTY data presented in this manuscript with the authors recently published TCR signaling interactomes from CD4 T cells (<https://www.ncbi.nlm.nih.gov/pubmed/31591574>). If the authors could analyze the dynamics of their phosphosites in respect to their presence in distinct protein complexes they may come up with a refined model and predictions of how these sites could regulate the dynamics of protein complex formation in TCR signaling. Can the authors do this analysis at least for the discussed role of ITSN2 in TCR internalization?

We appreciate the interest of Reviewer #2 for our studies. We fully agree, this is a very good suggestion that we actually started to explore but that requires further algorithm development. In particular it is difficult to apply such analysis with ITSN2 and CD3 chains since none of these two molecules were a bait in the mentioned study (Voisinne et al 2019). As indicated in the discussion, our interactomic study indicated that ITSN2 constitutively interacts with CBL and in a TCR-inducible manner with CBL-B. However, as reported by previous studies, these interactions could involve proline rich sequences of CBLs and the SH3 domains of ITSNs rather than phosphorylation mechanisms (Martin and al, Mol. Pharmacol 2006; Nikolaienko and al, 2009 Biopolym. Cell). Thus, further analyses are needed to elucidate the molecular mechanism associated with the ITSN2 functions. This is currently out of the scope of this manuscript but the combined exploitation of dynamic of phosphosites with protein interactions will certainly be considered in future analyses.

- The correlation of phosphosite occupancy with nuclear/cytoplasm ratio of the protein is validated for NFATC2 and FOXO3. It would be better to approach this important functional aspect of the reported phosphosites globally and evaluate nuclear/cytoplasm ratios after stimulation not by WB as shown in Fig EV3, but rather by a proteomic approach. Can the authors perform mass spec on samples prepared like in Fig EV4c and correlate the site occupancy in their pSTY datasets with nuclear/cytoplasm ratios for all proteins? This would be a valuable addition to the dataset.

We thank Reviewer #2 for this suggestion. Under the present conditions related to the SARS-CoV-2 virus it is impossible to perform these experiments. When we will have the possibility to resume our work both in Marseille and Toulouse laboratories this suggestion will be considered.

Minor comments:

- Figure EV1c: labelling of y axis is missing

This omission has been fixed.

- Numbers in the text of the unique phosphorylation sites don't correspond to the numbers in Figure EV1b

The numbers of EV1B are related to all classes (1, 2, 3 and 4) of localization scores (before filtering), see legend:

“** number of phosphorylated sites (or combination of sites in the case of multiply phosphorylated peptides) determined from MaxQuant «Phospho (STY)Sites.txt» tables. Class 1, 2, 3 and 4 correspond to a phospho-localization score of >75%; 50% <= 75%; 25% <= 50%, and <= 25%, respectively.”

while the numbers in the initial version of text corresponded only to phosphorylation sites with high confidence localization (see Material and Methods).

The text has been modified as follows (page 5), in order to describe more explicitly the results shown in each figure or Table:

“These led to the identification of 13,009 unique phosphorylated peptides corresponding to 9,702 and 560 phospho-sites –or combination of sites in the case of multiple phosphorylated peptides– in the TiO2 and the pY-IP data set, respectively (Figure EV1b). From these peptides, we were able after filtering (PEP value > 0.01, localization score >75%, elimination of sites with high number of missing values, see Material and Methods) to relatively quantify 7,180 phosphorylation sites or combinations of sites (Dataset EV1). These correspond to 6,984 unique, non-redundant sites (Figure 1b-c), including 5659 phospho-serines (pS), 1,070 phospho-threonines (pT) and 254 pY present on 2,118 proteins in primary T cells”.

- Figure EV2: figure captions for EV2a and EV2b are mixed-up

We thank Reviewer #2 for having noticed this point. The figure legend has been fixed in a modified version of the manuscript.

- Figure 4e: total ERK1/2 is missing in the WB

The same membrane was used for all the proteins displayed. Accordingly, we re-probed the membrane with anti-ACACA and anti-AMPKa1 antibodies for loading control (like for the phospho-ERK1/2 immunoblot). We apologize for this not being clear in the manuscript.

Reviewer #3:

In this manuscript, Locard-Paulet et al provide a rigorous analysis of the phosphoproteome of CD4 T cells stimulated by antibody or MHC tetramer through multiple timepoints. Although others have attempted to characterize the primary T cell phosphoproteome previously, those studies lacked the rigor and in-depth computational analysis to truly exploit this type of data. This study is unique both in the number of timepoints examined as well as the excellent follow-up study on the intersectin 2 protein that was uncovered. Showing that wide-scale analysis uncovered the modulation of the T cell activation threshold by ITSN2 was an excellent proof of concept that this data has great utility. Overall, this study provides the most rigorous view to date of the T cell phosphoproteome in primary T cells and provides modes of analysis of the large wealth of quantitative phosphorylation information that not only reveal new signaling pathways in T cells but set the bar for future studies beyond T cells to extract biological meaning out of these sorts of wide-scale experiments. This is a fantastic window into T cell signaling.

We thank Reviewer #3 for summarizing and highlighting the strengths of our work.

Major Questions:

1. Primary T cell phosphoproteomics has been performed previously and the methods employed to make the phosphoproteomic measurements in this study are not intended to be novel. The introduction in particular should compare and contrast the current work with the previous efforts. In particular, the 2017 Front. Immunol paper by Joshi et al and the 2012 article in MCP by Ruperez et al and the 2014 article by Navarro et al in MCP should be compared and contrasted with the current study. The Joshi study in particular did compare naive to CD3/28 stimulated primary human T cells and included a rigorous number of replicates and a large number of sites identified. The Ruperez et al study also looked at both the total and pY phosphoproteome, very similarly to this study. I think there is a very strong argument that this study transcends these others but I feel like it needs to be made more explicitly to highlight the novelty. The discussion mentions that this manuscript focuses "for the first time" on early phosphorylation events but the Joshi et al paper published in 2017 did compare 5 minute CD3/28 stimulated to naive human CD4 T cells. The authors also mention in the introduction "despite these efforts, a global understanding of this complex signaling network is still lacking". Could you please be more specific on how the other primary T cell phosphoproteomics studies have fallen short to showcase the novelty of this study. I would say that the computational analysis of your study transcends anything done previously and transforms the study into a more generally useful resource. Also consider rewording "we have developed" on page 4 to indicate that you are using biochemical methods that others have

developed i.e. TiO₂/ pTyr100 IP and their combination (see Ruperez et al and numerous other papers that have used the same methods). Certainly, this is not a method development paper where phosphoproteomic methods are being developed and optimized.

We thank Reviewer #3 for advising us on improving the form of the manuscript. We understand the concerns raised by Reviewer #3 and have modified the introduction of the manuscript based on his/her suggestions.

“Such large-scale MS-based approaches have been already used in a few studies to characterize protein phosphorylation and associated molecular mechanisms in primary T cells. Navarro et al {Navarro, 2011 #576} applied SILAC protein metabolic labeling on murine P14 cytotoxic T lymphocytes (CTLs) to analyze phosphorylation following long-term (1h) stimulation of the TCR with its cognate peptide, and identified around 2,000 phosphopeptides, among which 22% were TCR-regulated. Subsequently, to study the mechanisms of PKD2, a kinase important for effector cytokine production after TCR engagement, a similar strategy was implemented to compare the phosphoproteomes of wild type and PKD2 deficient CTLs after 5min of TCR activation {Navarro, 2014 #1956}. This work led to a more extensive coverage of 15,000 site-specific phosphorylations in antigen receptor activated CTLs, although no comparison was performed with unstimulated cells, precluding the identification of TCR-regulated phosphorylation sites. In a recent study, Joshi et al also used phosphoproteomics to analyze regulatory T cells (Treg) suppression mechanisms on primary human conventional T cells, upon TCR stimulation and Treg-mediated suppression, respectively {Joshi, 2017 #1957}. Using a coculture system and a quantitative approach based on isotopic dimethyl labeling of peptides, the authors could detect around 2,000 phosphopeptides and quantify around 1,000 of them in three different T-cell states (unstimulated, TCR-stimulated with anti-CD3/anti-CD28 antibodies, and Treg-suppressed). These studies, based either on metabolic protein labeling or on dimethyl peptide labeling, were limited in the number of conditions or time points that could be included and compared. In addition, they focused on the global phosphoproteome, composed mainly of phosphorylated serine and threonine sites. To overcome this limitation, Ruperez et al introduced an additional step of purification to specifically enrich phosphorylated tyrosine residues {Ruperez, 2012 #634}. Using this approach, the authors identified a total of 2,883 phosphorylated peptides in CD4 human T cells stimulated for 5min with anti-CD3 antibodies, including 48 peptides phosphorylated on tyrosine. All together, these developed methods have paved the way for in-depth analysis of signaling and phosphorylation induced during T cell activation.”

2. Unlike most other studies, this study also runs the control experiment to test whether protein abundance has changed significantly to explain a subset of the phosphorylation changes. The statement is made on page 6 that "protein abundances were not impacted by TCR activation at these early timepoints" but surely only some fraction of the proteins with observed phosphorylation sites were detected in the total proteome analysis as only 6500 proteins were quantified according to figure EV2. Looks like maybe 10-20% were not quantified according to EV2. Could you please report this percentage in the text of the manuscript and indicate that this information for individual proteins is available in your web tool. The concern is that a T cell signaling researcher might be interested in a particular site and not know to check for the unphosphorylated protein data or to collect western blot data on this protein if the particular protein was not observed. For example, for the sites featured in Figure 2c, could you mark differently the sites on proteins that were not captured in the

unphopsho dataset or if they were all captured, state that?

We thank Reviewer #3 for raising this concern, and we agree that looking at protein relative abundances is an important control for such phospho-signaling analysis. We have amended the text of the manuscript accordingly as follows (page 7):

“A similar strategy applied to the data obtained from the analysis of the peptides before phospho-enrichment allowed us to quantify more than 80% of the abundance of regulated phospho-proteins. Importantly, these data indicate that protein abundances were not impacted by TCR activation at these early time points (Figure 1a, Figure EV2b and Dataset EV2), and that changes of phospho-peptide quantities directly reflected phosphorylation or dephosphorylation events”

The individual information whether the proteins have been quantified in the proteome and if they are regulated according to our statistical thresholds are listed in Dataset EV1 (column “WarningProteinFC”) and Dataset EV2 (column “Regulation”).

Regarding Figure 2C, none of the phospho-proteins displayed show significant changes in abundances after TCR stimulation, and the only proteins that were not quantified in our proteome are Raf1 and Braf.

3. One of the most interesting parts of your data was the relatively slow kinetics of ITAM CD3 subunit and zeta phosphorylation relative to phosphorylation sites typically believed to be downstream of ITAM phosphorylation. For example, could you please discuss why Erk, PLCg1, Lck, and Zap70 phosphorylation appear to happen much faster than ITAM phosphorylation. Do you suspect that ITAMs are not leading to phosphorylation of these sites in your cells?

We thank Reviewer #3 for his/her constructive comment very similar to the one raised by Reviewer #1.

Here is the common addressed response:

We understand that based on the unique cluster numeration, one could conclude that receptor phosphorylation occurs after phosphorylation of the proteins localized in C1-4. This is actually not the case. In fact, the clustering is really based on the shape of the kinetics, it is performed after scaling of the data and thus, it doesn't take the amplitude into account. The induction of the phosphorylation of CD3 chains is actually very rapid and is already intense at 15 seconds but not yet maximum (see figure below). This phosphorylation increases progressively during the stimulation and reaches a maximum between 300 and 600 seconds. In contrast, the phosphorylation of proteins in C1-C4 reach their maximum very rapidly (typically at 15 or 30s) and some are very transient. For example, phosphorylation at position 185 of Mapk1, a key effector downstream of the receptor, is maximal at 30s and is downregulated afterwards (cluster 2). As illustrated below, the overlay of phosphorylation dynamics of CD3-Y142 and MAPK1-Y185 shows that their inductions and amplitudes are similar (the intensity has been normalized on the NS condition).

We wrote in the text that clusters C5-C8 had a “slower dynamic”, which is maybe misleading, In the revised version, we qualified the phosphorylation profiles as “transient” (clusters 1-2), “sustained” (3-6), or “progressive” (7-8).

The description of clusters 7 and 8 is now the following (page 9):

“For C7 and C8, we observed a gradual and continuous amplification of the phosphorylation signal all along the time-course, which remained sustained (C7) or was even increasing (C8) at the later time points”

Regarding the progressive phosphorylation of the TCR, one explanation could be that the phosphorylation of only a few CD3 chains is sufficient to trigger a large wave of phosphorylation of downstream proteins (C1-C4) (Huang, J. et al. *Immunity* 39, 846–857, 2013). The subsequent and progressive phosphorylation increase of the CD3 chains could be the result of the positive feedback mechanism involving the stabilization of the active conformation of Lck promoting a progressive increase of the ZAP70 kinase activity combined with the protection of phosphorylated ITAMs from surrounding phosphatases by the binding of ZAP70. In this scenario, ITAM phosphorylation could stay high even though the phosphorylation of downstream effectors drops. In a non-exclusive manner, this phenomenon could also reflect a progressive accumulation of endocytosed CD3 chains that remain phosphorylated intracellularly.

As mentioned in the discussion of the manuscript, this observation remains fully compatible with models of TCR activation.

Also, the way the data is presented as cluster patterns, it is hard to tell which timepoints are significantly changed. I assume that the requirement for inclusion in this figure is that at least one but not necessarily all timepoints must be significantly altered. A supplemental heatmap

figure for data in fig 2c showing exactly which timepoints are significant relative to the 0 min point would be helpful.

The reviewer is right, as stated in the Materials and Methods (page 23), sites were considered statistically significant (regulated) if they presented a p-value of the ANOVA ≤ 0.05 and a minimum of one couple of time points with more than 2 points per condition and a corrected p-value ≤ 0.05 with an associated absolute fold change > 1.75 . Thus, it is possible that only one couple of time point (usually including the 0min time point) would present a significant difference of signal. In the previous version of **Dataset EV1**, the column “BestTimePoint” indicated the couple of time points with the highest fold change among those that were significantly different. In addition, for all the possible pairwise comparisons of time points, it included a columns displaying the “fold change” and “Tukey corrected p-value”. In the present version, we added additional columns that clearly indicates, for each of these pairwise comparisons of time points, if the site is actually considered regulated according to the previously selected threshold values (indicated as “TRUE” and colored in red).

In addition, we provide as suggested by the reviewer, a list for the subset of sites shown in Figure 2C, as an additional sheet in Dataset EV1.

Minor Questions:

1. page 3, not sure "signalization cascades" is a word.

This has been changed by “signaling cascades”

2. I was a bit confused when you mentioned at the bottom of page 10 the phosphorylation of putative Erk and Akt sites on STMN1 and other proteins. Are these predictions of the direct phosphorylation of these proteins in general by ERK/Akt in phosphosite or sites that show an Erk/Akt motif. Are these Erk/Akt pathway or direct targets? Are the interactions direct or indirect? Maybe you could explain in a bit more detail the phosphosite kinase-substrate relationship data that you are using here.

The analysis mentioned by Reviewer #3 is not based on predictions but was based on kinase substrate pairs reported in the literature and listed in the PhosphoSitePlus database. This has now been clarified in the text (page 11). In the example chosen by Reviewer #3, Stmn1-S63 and Stmn1-S25 have been shown to be substrate of ERK1/2 (*J Biol Chem* 268, 25671-80, *J Biol Chem* 273, 22848-55, *Neuroscience* 219, 255-70, *Proteomics* 9, 4495-506) while there is no report that indicates their phosphorylation by AKT. Determining whether phosphorylation and interactions between kinases and substrates are direct, is always difficult to assess. In the cited studies no direct evidence is provided to support these statements. Only kinase assays in which substrates and kinases have been generated synthetically in vitro could address these issues.

3. Not really a question but a comment that the degree of overlap of the OT-I and anti-CD3/4 dataset was really a strong supporter of the quality of your data. In particular supplemental figure EV4b-d and especially EV4d was very powerful and you may consider upgrading it into the regular figures from the supplement. I am not aware that tetramer stimulation

phosphoproteomic analysis has been performed previously in primary T cells and you might want to hype this up more in your intro.

We appreciate the interest of Reviewer #3 for this part of the manuscript. We are currently developing this method of stimulation to address important issues of T cell activation associated with the capacity of T cells to discriminate specific antigens. These ongoing analyses will be the object of a complete study.

4. For Figure EV4b, consider switching to a pseudocolor dot figure to indicate the density instead of contour plot although this might just be a point of preference.

We thank Reviewer #3 for his/her suggestion. This request has been done in a new version of Figure EV4.

15th May 2020

Manuscript Number: MSB-20-9524R

Title: LymphoAtlas: a dynamic and integrated phosphoproteomic resource of TCR signaling in primary T cells

Author: Marie Locard-Paulet

Guillaume Voisinne

Carine Froment

Marisa Goncalves Menoita

Youcef Ounoughene

Laura GIRARD

Claude Gregoire

Daiki Mori

Manuel Martinez

Hervé Luche

Jérôme Garin

Marie Malissen

Odile Schiltz

Bernard Malissen

Anne Gonzalez de Peredo

Romain Roncagalli

Dear Dr Roncagalli,

Thank you for sending us your revised manuscript. We have now heard back from the three reviewers who agreed to evaluate your manuscript. You will see from the comments below that the reviewers are now satisfied with the revision and support publication of the article in *Molecular Systems Biology*. I am pleased to inform you that your manuscript will be accepted in principle pending the following essential amendments:

- Please provide up to 5 keywords and incorporate them in the main text.

-Dataset EV2 needs a title added in its legend in the file.

-I notice that you have deposited the proteomics data to ProteomeXchange. Please make sure that the datasets are made publicly available upon acceptance.

- Please provide a "standfirst text" summarizing the study in one or two sentences (approximately 250 characters, including space), three to four "bullet points" highlighting the main findings and a "synopsis image" (550px width and max 400px height, jpeg format) to highlight the paper on our homepage.

Here are a couple of examples:

<https://www.embopress.org/doi/10.15252/msb.20199195>

<https://www.embopress.org/doi/10.15252/msb.20199170>

<https://www.embopress.org/doi/10.15252/msb.20199265>

-Our data editor has made a couple of suggestions to your manuscript (see attached word file),

please address these issues.

When you resubmit your manuscript, please download our CHECKLIST (<http://bit.ly/EMBOPressAuthorChecklist>) and include the completed form in your submission. *Please note* that the Author Checklist will be published alongside the paper as part of the transparent process (<https://www.embopress.org/page/journal/17444292/authorguide#transparentprocess>)

Click on the link below to submit your revised paper.

Yours sincerely,

Jingyi Hou
Editor
Molecular Systems Biology

If you do choose to resubmit, please click on the link below to submit the revision online before 14th Jun 2020.

IMPORTANT: When you send your revision, we will require the following items:

1. the manuscript text in LaTeX, RTF or MS Word format
2. a letter with a detailed description of the changes made in response to the referees. Please specify clearly the exact places in the text (pages and paragraphs) where each change has been made in response to each specific comment given
3. three to four 'bullet points' highlighting the main findings of your study
4. a short 'blurb' text summarizing in two sentences the study (max. 250 characters)
5. a 'thumbnail image' (550px width and max 400px height, Illustrator, PowerPoint or jpeg format), which can be used as 'visual title' for the synopsis section of your paper.
6. Please include an author contributions statement after the Acknowledgements section (see <https://www.embopress.org/page/journal/17444292/authorguide#manuscriptpreparation>)
7. Please complete the CHECKLIST available at (<http://bit.ly/EMBOPressAuthorChecklist>). Please note that the Author Checklist will be published alongside the paper as part of the transparent process (<https://www.embopress.org/page/journal/17444292/authorguide#transparentprocess>).
8. Please note that corresponding authors are required to supply an ORCID ID for their name upon submission of a revised manuscript (EMBO Press signed a joint statement to encourage ORCID adoption) (<https://www.embopress.org/page/journal/17444292/authorguide#editorialprocess>).

Currently, our records indicate that the ORCID for your account is 0000-0001-7554-0552.

Link Not Available

The system will prompt you to fill in your funding and payment information. This will allow Wiley to send you a quote for the article processing charge (APC) in case of acceptance. This quote takes into account any reduction or fee waivers that you may be eligible for. Authors do not need to pay any fees before their manuscript is accepted and transferred to the publisher.

REFEREE REPORTS

Reviewer #1:

All comments have been addressed and manuscript can be published

Reviewer #2:

The authors have adequately responded to my concerns. In the current situation I understand that collecting further experimental evidence is not possible. I recommend publication of this work.

Reviewer #3:

In this manuscript, Locard-Paulet et al provide a rigorous analysis of the phosphoproteome of CD4 T cells stimulated by antibody or MHC tetramer through multiple timepoints. Although others have attempted to characterize the primary T cell phosphoproteome previously, those studies lacked the rigor and in-depth computational analysis to truly exploit this type of data. This study is unique both in the number of timepoints examined as well as the excellent follow-up study on the intersectin 2 protein that was uncovered. Showing that wide-scale analysis uncovered the modulation of the T cell activation threshold by ITSN2 was an excellent proof of concept that this data has great utility. Overall, this study provides the most rigorous view to date of the T cell phosphoproteome in primary T cells and provides modes of analysis of the large wealth of quantitative phosphorylation information that not only reveal new signaling pathways in T cells but set the bar for future studies beyond T cells to extract biological meaning out of these sorts of wide-scale experiments. This is a fantastic window into T cell signaling. This newly revised manuscript has thoughtfully and comprehensively addressed all of the three reviewer's comments. There is no doubt that this study

will have lasting impact on the T cell signaling community by providing a compendium of knowledge about the rapid phosphorylation changes triggered by TCR crosslink.

The Authors have made the requested editorial changes.

YOU MUST COMPLETE ALL CELLS WITH A PINK BACKGROUND ↓
 PLEASE NOTE THAT THIS CHECKLIST WILL BE PUBLISHED ALONGSIDE YOUR PAPER

Corresponding Author Name: Roncagalli Romain
 Journal Submitted to: Molecular Systems Biology
 Manuscript Number: MSB-20-9524